# Biomimetic cardiac tissue culture model (CTCM) to emulate cardiac physiology and pathophysiology ex vivo

Jessica M. Miller[1,2,8], Moustafa H. Meki[1,2,8], Ahmed Elnakib[2], Qinghui Ou[1], Riham R. E. Abouleisa[1], Xian-Liang Tang[1], Abou Bakr M. Salama[1,3], Ahmad Gebreil[1], Cindy Lin[1], Hisham Abdeltawab[2], Fahmi Khalifa[2], Bradford G. Hill[4], Najah Abi-Gerges[5], Roberto Bolli[1], Ayman S. El-Baz[2], Guruprasad A. Giridharan[2] & Tamer M. A. Mohamed [1,2,4,6,7✉]

There is need for a reliable in vitro system that can accurately replicate the cardiac physiological environment for drug testing. The limited availability of human heart tissue culture systems has led to inaccurate interpretations of cardiac-related drug effects. Here, we developed a cardiac tissue culture model (CTCM) that can electro-mechanically stimulate heart slices with physiological stretches in systole and diastole during the cardiac cycle. After 12 days in culture, this approach partially improved the viability of heart slices but did not completely maintain their structural integrity. Therefore, following small molecule screening, we found that the incorporation of 100 nM tri-iodothyronine (T3) and 1 μM dexamethasone (Dex) into our culture media preserved the microscopic structure of the slices for 12 days. When combined with T3/Dex treatment, the CTCM system maintained the transcriptional profile, viability, metabolic activity, and structural integrity for 12 days at the same levels as the fresh heart tissue. Furthermore, overstretching the cardiac tissue induced cardiac hypertrophic signaling in culture, which provides a proof of concept for the ability of the CTCM to emulate cardiac stretch-induced hypertrophic conditions. In conclusion, CTCM can emulate cardiac physiology and pathophysiology in culture for an extended time, thereby enabling reliable drug screening.

[1] From the Institute of Molecular Cardiology, Department of Medicine, University of Louisville, Louisville, USA. [2] Department of Bioengineering, University of Louisville, Louisville, USA. [3] Faculty of Medicine, Zagazig University, Zagazig, Egypt. [4] Envirome Institute, Diabetes and Obesity Center, Department of Medicine, University of Louisville, Louisville, USA. [5] AnaBios Corporation, San Diego, USA. [6] Department of Pharmacology and Toxicology, University of Louisville, Louisville, USA. [7] Institute of Cardiovascular Sciences, University of Manchester, Manchester, United Kingdom. [8] These authors contributed equally: Jessica M. Miller, Moustafa H. Meki. ✉email: tamer.mohamed@louisville.edu

Prior to clinical studies, there is a need for reliable in vitro systems that could accurately replicate the human heart's physiological environment. Such systems should model altered mechanical stretches, heart rate, and electrophysiological properties. Animal models, the prevalently used cardiac physiology screening platform, have limited reliability in mirroring the effects of drugs seen in human hearts[1,2]. Ultimately, the ideal experimental cardiac tissue culture model (CTCM) is the one that demonstrates high sensitivity and specificity for various therapeutic and pharmacological interventions while accurately replicating the physiology and pathophysiology of the human heart[3]. The lack of such a system has limited drug discovery for new heart failure therapeutics[4,5] and resulted in drug-induced cardiotoxicity being a major cause of market withdrawal[6].

In the last decade, eight non-cardiovascular drugs have been withdrawn from clinical use because they induce QT interval prolongation, resulting in ventricular arrhythmia and sudden death[7]. Therefore, there is a growing need for reliable preclinical screening strategies to assess cardiovascular efficacy and toxicity. The recent shift toward the use of human-induced pluripotent stem cell-derived cardiomyocytes (hiPS-CMs) in drug screening and toxicity testing has provided a partial solution to this issue; however, the immature nature of the hiPS-CMs and the lack of the multicellular complexity of the heart tissue are major limitations of this technology[8]. Recent efforts have shown that this limitation can be partially overcome if early-stage hiPS-CMs, soon after the initiation of spontaneous contractions, are used to form cardiac tissue hydrogels and are subjected to a gradual increase in electrical stimulation over time[9]. However, these hiPS-CMs microtissues lack the mature electrophysiological and contractile properties of the adult human myocardium. Moreover, the human heart tissue is structurally more complicated, composed of a heterogeneous mixture of various cell types, including endothelial cells, neurons, and stromal fibroblasts linked together with a particular array of extracellular matrix proteins[10]. This heterogeneity of the non-cardiomyocyte cell population[11-13] in the adult mammalian heart is a major obstacle in modeling heart tissue using individual cell types. These major limitations highlight the importance of developing methods to culture intact myocardial tissue under physiological and pathological conditions[9].

Cultured thin (300 μm) human heart slices prove to be a promising model of the intact human myocardium. This technology provides access to a complete three-dimensional multicellular system similar to the human heart tissue. However, before 2019, culturing heart slices usage was limited by the short period of viability in culture (24 h)[14-16]. This is due to multiple factors, including the lack of physiologic mechanical stretching, the air-liquid interface, and the use of a simple culture medium that does not support the demands of the cardiac tissue. In 2019, several groups demonstrated that the incorporation of mechanical factors into cardiac tissue culture systems could extend the culture life, improve cardiac expression, and model cardiac pathology. Two elegant studies[17], and[18], have shown the positive influence of uniaxial mechanical loading on the cardiac phenotype during culture. However, these studies did not use the dynamic three-dimensional physiological mechanical loading of the cardiac cycle since heart slices were either loaded with isometric stretch forces[17] or linear auxotonic loading[18]. These methods of stretching the tissue resulted in the downregulation of multiple cardiac genes or the overexpression of genes related to pathological stretch response. Notably, Pitoulis et al.[19] developed a dynamic heart slice culture bath to recreate the cardiac cycle using force transducer feedback and a stretcher actuator. While this system allowed for the cardiac cycle to be more accurately simulated in vitro, the method's complexity and low throughput limits the application of the system. Our lab has recently developed a simplified culture system using electrical stimulation and an optimized culture medium to keep pig and human cardiac tissue slices viable for up to 6 days[20,21].

In the current manuscript, using porcine heart slices, we describe a cardiac tissue culture model (CTCM) that recapitulates the three-dimensional cardiac physiological and pathophysiological stretches during the cardiac cycle combined with humoral cues. This CTCM has the potential to advance the accuracy of pre-clinical drug prediction to previously unattainable levels by providing a medium-throughput, cost-effective heart system that mimics mammalian cardiac physiological/pathophysiological stretches for preclinical drug testing.

## Results

### CTCM maintains heart tissue physiological stretch for 12 days in culture

Hemodynamic mechanical cues play a critical role in preserving the functionality of cardiomyocytes in vitro[22-24]. In the current manuscript, we developed a CTCM (Fig. 1a) that can emulate the adult cardiac milieu by inducing simultaneous electrical and mechanical stimulation at the physiological frequency (1.2 Hz, 72 beats per minute). To avoid over-stretching the tissue during the diastolic phase, the tissues were oversized by 25% using a three dimensionally printed apparatus (Fig. 1b). Electrical stimulation induced by a C-PACE system was synchronized to initiate at 100 ms before the systolic phase using a data acquisition system to fully reproduce the cardiac cycle. The tissue culture system uses a programmable pneumatic driver (LB engineering, Germany) to cyclically distend a flexible silicone membrane inducing the stretch of the heart slices in the chambers above. The system is connected to an external air line with a pressure probe allowing fine pressure tuning (±1 mmHg) and timing (±1 ms) (Fig. 1c).

Using one pneumatic driver, we can operate 4 CTCM devices, each accommodating 6 tissue slices (Fig. 1d). Within the CTCM, the air pressures in the air chambers are translated into synchronized pressures in the fluid chamber and produce physiological stretch of the heart slices (Fig. 2a and Supplementary movie 1). Assessment of the tissue stretch at 80 mmHg resulted in a 25% stretch of the tissue slice (Fig. 2b). This percent stretch has been shown to correspond with a physiological sarcomere length, 2.2–2.3 μm, for normal contractility of heart slices[17,19,25]. Tissue movements are assessed using a custom camera setup (Supplementary Fig. 1). The tissue movement amplitude and the speed rate (Fig. 2c, d) are consistent with the stretching during the cardiac cycle and the timing during systole and diastole (Fig. 2b). The heart tissue stretch and speed during contraction and relaxation remained consistent over the 12 days in culture (Fig. 2e). To assess the effect of electrical stimulation on contractility during culture, we developed a method to determine the live strain using the Shape from Shading algorithm (Supplementary Fig. 2a, b) and were able to differentiate between the strain with and without electrical stimulation from the same heart slices (Fig. 2f). The strain with electrical stimulation is 20% higher than without electrical stimulation within the moving regions of the slices (R6-9), which indicates the contribution of the electrical stimulation to the contractile function.

### Heart slices cultured in CTCM maintain entire viability but not structural integrity over 12 days

In our previous static biomimetic heart slice culture system[20,21], we maintained the viability, functionality, and structural integrity of heart slices for 6 days with the application of electrical stimulation and optimized composition of the culture medium. However, after 10 days there was a sharp decline in these parameters. We will refer to slices

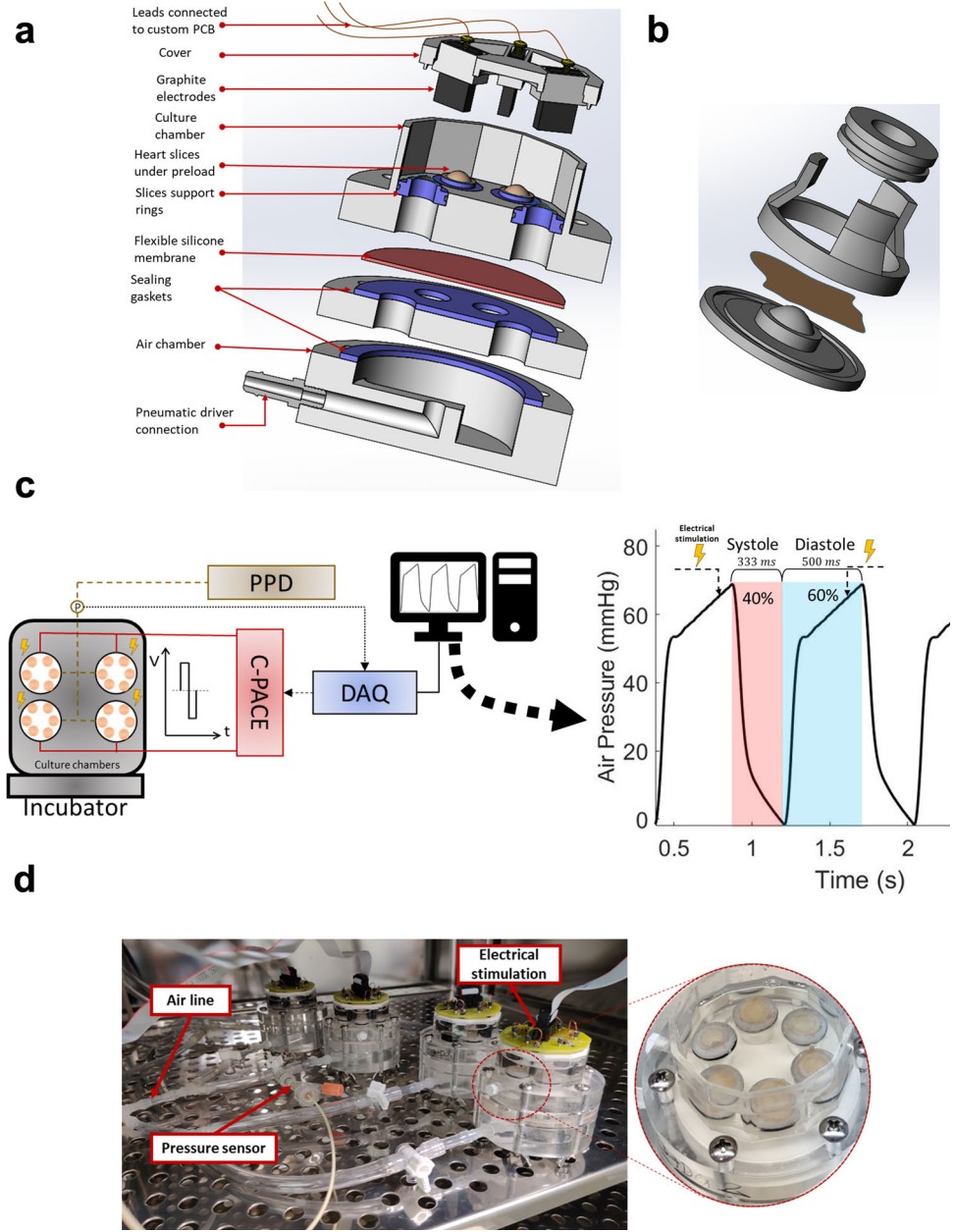

**Fig. 1 Illustration of the cardiac tissue culture model (CTCM). a** The tissue slices are attached to 7-mm diameter support rings shown in blue within the culture chamber of the device. The culture chamber was separated from the air chamber by a thin, flexible silicone membrane. Between each chamber, a sealing gasket was placed to prevent leaks. The cover for the device contains graphite electrodes that provide electrical stimulation. **b** Schematic illustration of tissue oversizing apparatus, ring guide, and support ring. Tissue slices (brown) are positioned on the oversizing device, and the ring guide is placed in the grove on the outer edge of the apparatus. Using the guide, the support ring coated with histoacryl glue is carefully placed on the heart tissue slice. **c** A diagram depicting the timing of the electrical stimulation in relation to the pressure within the air chamber controlled by the programmable pneumatic driver (PPD). A data acquisition device was used to synchronize electrical stimulation using a pressure probe sensor. When the pressure within the culture chamber reaches a specified threshold, an impulse signal is sent to the C-PACE-EM device to induce electrical stimulation. **d** Image of four CTCM devices set up on a shelf of an incubator. These four devices are connected to a single PPD by airlines, and a pressure sensor is inserted into a hemostatic valve to monitor the pressure within the airline. Each device can accommodate six tissue slices.

cultured in our previous static biomimetic culture system[20,21] as a control condition (Ctrl), and we will refer to the slices cultured under synchronized mechanical and electrical stimulation (CTCM) using our previously optimized culture media as MC condition. First, we determined that the mechanical stimulation without electrical stimulation is insufficient to maintain tissue viability for 6 days (Supplementary Fig. 3a, b). Interestingly, by introducing both physiological mechanical, and electrical stimulations using the CTCM, the viability of 12-day heart slices was

maintained similar to fresh heart slices in the MC condition but not in the Ctrl condition, as shown by the MTT assay (Fig. 3a). This indicates that the mechanical stimulation and simulation of the cardiac cycle can maintain the viability of tissue slices for twice the time reported in our previous static culture system. However, assessment of the structural integrity of the tissue slices by immunolabeling for cardiac Troponin-T and Connexin 43 demonstrated that connexin 43 expression of day 12 MC tissue was significantly higher than the same day control. Still, the

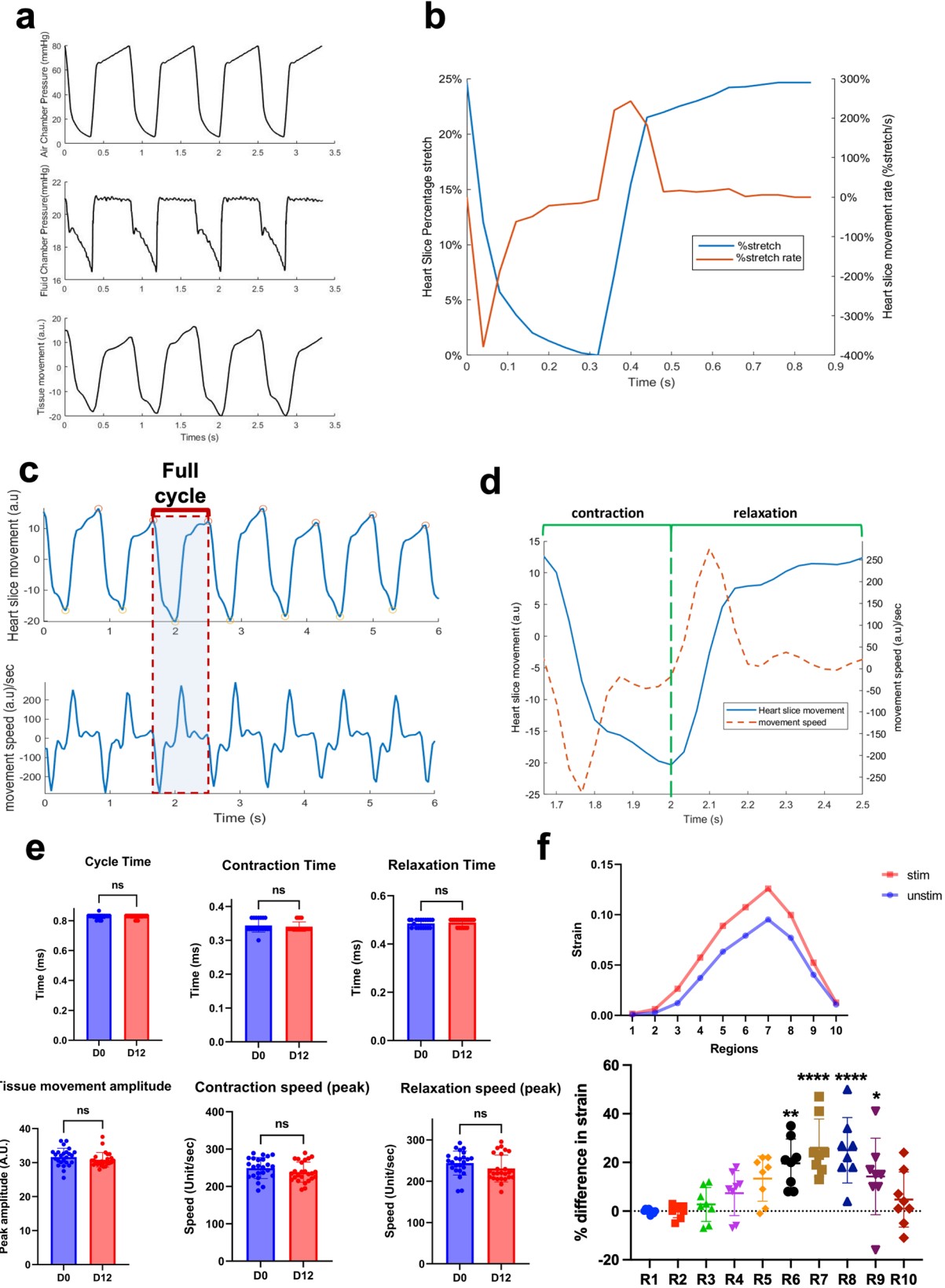

uniform expression of connexin43 and Z-disc formation was not fully maintained (Fig. 3b). We used an artificial intelligence (AI) based framework to quantify tissue structural integrity[26], based on an image-based deep learning pipeline for the automated quantification of the structural integrity of the heart slices based on troponin-T and Connexin 43 staining in terms of localization and fluorescent intensity. This technique uses a convolutional neural network (CNN) and the deep learning framework to reliably quantify the cardiac tissue structural integrity in an automated and unbiased manner, as described in ref. [26]. MC tissue showed an improved structural similarity to day 0 compared to the static control slices. Furthermore, Masson's

**Fig. 2 Characterization of the CTCM and evaluation of the heart slice stretches over time in culture. a** Representative traces of the air chamber pressure, fluid chamber pressure, and tissue movement measurements verified that the air chamber pressure changes the fluid chamber pressure, which induces a corresponding tissue slice movement. **b** Representative traces of the percent stretch (blue) of the tissue slices correspond with the percent stretch rate (orange). **c** The heart slice movement measured was consistent with the measured movement speed. (**d**) Representative trace of the heart slice cycle movements (blue line) and speed (dotted orange line). **e** Quantification of the cycle time ($n = 19$ slices/group from different pigs), contraction time ($n = 19$ slices/group), relaxation time ($n = 19$ slices/group from different pigs), tissue movement ($n = 25$ slices/group from different pigs), peak contraction speed ($n = 24$ (D0), 25 (D12) slices/group from different pigs); and peak relaxation speeds ($n = 24$(D0), 25 (D12) slices/group from different pigs). Two-tailed Student $t$-tests revealed no significant difference in any of the parameters. **f** Representative trace for strain analysis of a tissue slice with (red) and without (blue) electrical stimulation over ten regional areas of the tissue slice from the same slice. The bottom panel shows the quantification of the percentage difference in the strain of tissue slices with and without electrical stimulation over ten regional areas from different slices. ($n = 8$ slices/ group from different pigs, Two-tailed Student $t$-test is performed; ****$p < 0.0001$, **$p < 0.01$, *$p < 0.05$). Error bars are representative of the Mean ± SD.

trichrome stain showed a significantly lower percent area of fibrosis with the MC condition compared to the control condition on day 12 of culture (Fig. 3c). While the CTCM did improve the viability of day 12 heart tissue slices to levels similar to fresh heart tissue, it did not significantly improve the structural integrity of the heart slices.

**Small molecule screening to improve the heart tissue viability and structural integrity in CTCM.** We hypothesized that by incorporating small molecules into the culture media, cardiomyocyte integrity could be improved, and the development of fibrosis during the CTCM culture could be reduced. Therefore, we performed a small molecule screening using our static control culture[20,21] because of the lower number of confounding factors associated with it. Dexamethasone (Dex), tri-iodothyronine (T3), and SB431542 (SB) were selected for this screening. These small molecules have been previously used in hiPSC-CM cultures to induce cardiomyocyte maturation by improving sarcomere length, t-tubules, and conduction velocity[27,28]. In addition, both Dex, a glucocorticoid, and SB are known to suppress inflammation[29,30]. Therefore, we tested whether incorporating one or a combination of these small molecules would improve the structural integrity of the heart slices. For the initial screening, the dosage of each compound was selected based on the concentration typically used in cell culture models (1 μM Dex[27], 100 nM T3[27], and 2.5 μM SB[31]). Following 12 days in culture, the combination of T3 and Dex resulted in the best cardiomyocyte structural integrity and the lowest fibrotic remodeling (Supplementary Figs. 4 and 5). In addition, using either half or double of these concentrations of T3 and Dex had detrimental effects compared to the normal concentrations (Supplementary Fig. 6a, b).

**Combining T3 and Dex with the CTCM fully maintained pig heart slices similar to fresh heart slices for 12 days.** Following the initial screening, we performed head-to-head comparisons of 4 culture conditions (Fig. 4a); Ctrl: heart slices cultured in our previously described static culture with our optimized culture media;[20,21] TD: Ctrl with T3 and Dex added to the culture media; MC: heart slices cultured in CTCM using our previously optimized culture media; and MT: CTCM with T3 and Dex added to the culture media. After 12 days in culture, the viability of MC and MT tissues was maintained similar to fresh tissue as assessed by the MTT assay (Fig. 4b). Interestingly, the addition of T3 and Dex to the transwell culture (TD) did not significantly improve viability compared to the Ctrl condition, implying the vital role of mechanical stimulation in maintaining heart slice viability.

A shift in metabolic reliance from fatty acid oxidation to glycolysis is a hallmark of cardiomyocyte dedifferentiation. Immature cardiomyocytes primarily utilize glucose for ATP production and have underdeveloped mitochondria with few cristae[5,32]. The glucose utilization assay demonstrated that under

MC and MT conditions, glucose utilization was similar to day 0 tissue (Fig. 4c). However, Ctrl samples showed a significant increase in glucose utilization compared to fresh tissue. This indicates that combining CTCM and T3/Dex improves tissue viability and preserves the metabolic phenotype of heart slices cultured for 12 days. Furthermore, strain analysis showed maintenance of strain levels in the MT and MC conditions similar to fresh heart tissue for 12 days (Fig. 4d).

To analyze the overall impact of the CTCM and T3/Dex on the global transcriptional landscape of the heart slice tissue, we performed RNAseq on heart slices from all four different culture conditions (Supplementary data 1). Interestingly, MT slices showed high transcriptional similarity to the fresh heart tissue, with only 16 out of 13,642 genes differentially expressed. However, as we demonstrated before[21], the Ctrl slices showed 1229 differentially expressed genes after 10–12 days in culture (Fig. 4e). These data were validated by qRT-PCR of cardiac and fibroblast genes (Supplementary Fig. 7a–c). Interestingly, the Ctrl slices showed downregulation of the cardiac and cell cycle genes and an upregulation of inflammatory gene programs. These data indicate that the dedifferentiation that normally occurs following long-term culture was completely attenuated under the MT condition (Supplementary Fig. 8a, b). A closer examination of the sarcomeric genes revealed that only the MT conditions preserved the sarcomeric (Fig. 4f) and ion channels (Supplementary Fig. 9) encoding genes from downregulation seen in Ctrl, TD, and MC conditions. These data indicate that with the combination of mechanical and humoral stimulation (T3/Dex), the transcriptome of heart slices could be maintained similar to fresh heart slices for 12 days in culture.

These transcriptional findings were confirmed by the fact that the structural integrity of the cardiomyocytes in heart slices was best preserved for 12 days in the MT condition, as shown by the intact and localized gap junction protein, connexin 43 (Fig. 5a). Furthermore, fibrosis in heart slices in MT condition was significantly reduced compared to Ctrl and was similar to the fresh heart slices (Fig. 5b). These data indicate that the combination of mechanical stimulation and T3/Dex treatment effectively preserved heart slice cardiac structure for an extended time in culture.

**Inducing cardiac hypertrophy through overstretching the tissue in the CTCM.** Lastly, the ability of the CTCM to model cardiac hypertrophy was assessed by increasing the cardiac tissue stretch amplitude. In the CTCM, the peak pressure in the air chamber was increased from 80 mmHg (normal stretch) to 140 mmHg (Fig. 6a). This corresponds to an increase in the stretch by 32% (Fig. 6b), which has been previously shown to be the appropriate percent stretch necessary for a heart slice to achieve a sarcomere length similar to that seen in hypertrophy[17,19,25]. The heart tissue stretch and speed during contraction and relaxation remained consistent over the six days

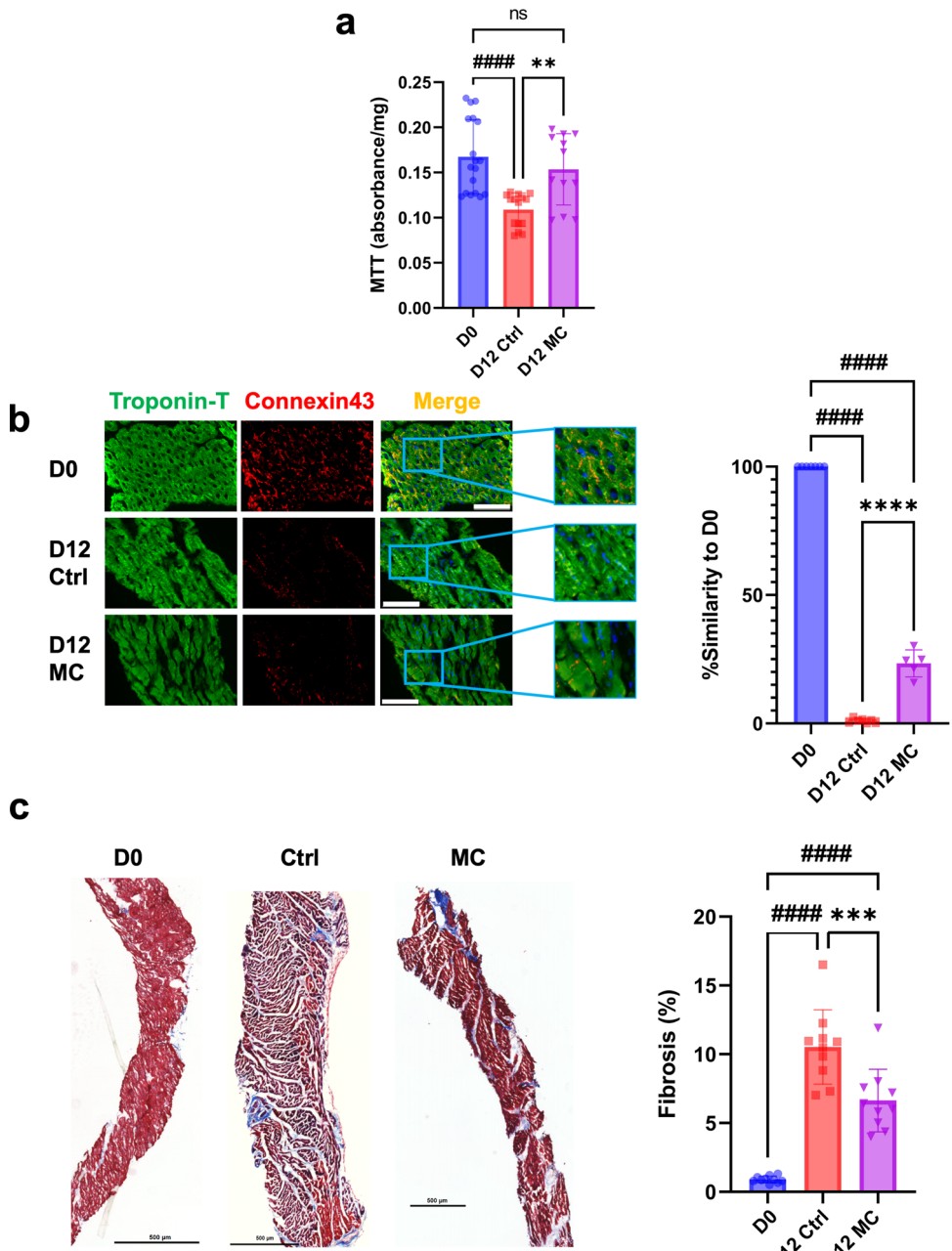

**Fig. 3 Mechanical stimulation partially improves tissue slice viability and structural integrity after 12 days in culture. a** Bar graph shows quantification of the MTT viability of fresh heart slices (D0) or heart slices culture for 12 days either in static culture (D12 Ctrl) or in CTCM (D12 MC) ($n = 18$ (D0), 15 (D12 Ctrl), 12 (D12 MC) slices/group from different pigs, one way ANOVA test is performed; ####$p < 0.0001$ compared to D0 and **$p < 0.01$ compared to D12 Ctrl). **b** Representative immunofluorescence images for troponin-T (green), connexin 43 (red), and DAPI (blue) for freshly isolated heart slices (D0) or heart slices cultured for 12 days under static conditions (Ctrl) or CTCM conditions (MC) (Scale bare = 100 μm). Artificial intelligence quantification of the heart tissue structural integrity ($n = 7$ (D0), 7 (D12 Ctrl), 5 (D12 MC) slices/group each from different pig, one-way ANOVA test is performed; ####$p < 0.0001$ compared to D0 and ****$p < 0.0001$ compared to D12 Ctrl). **c** Representative images (left) and quantification (right) for heart slices stained with Masson's trichrome stain (Scale bare = 500 μm) ($n = 10$ slices/group each from different pig, one-way ANOVA test is performed; ####$p < 0.0001$ compared to D0 and ***$p < 0.001$ compared to D12 Ctrl). Error bars are representative of the Mean ± SD.

of culture (Fig. 6c). Heart slices tissue from the MT conditions were either subjected to normal stretch (MT (Norm)) or over-stretching conditions (MT (OS)) for six days. As early as four days in culture, the hypertrophic biomarker, NT-ProBNP, was significantly increased in the culture media in MT (OS) conditions compared to the MT (Norm) conditions (Fig. 7a). Furthermore, following six days in culture, the cell size in MT (OS)

(Fig. 7b) was significantly increased compared to MT (Norm) heart slices. Furthermore, NFATC4 nuclear translocation was significantly increased in over-stretched tissues (Fig. 7c). These results show the progressive development of pathological remodeling following overstretching and provide a proof of concept that the CTCM device can be used as a platform to study stretch-induced cardiac hypertrophy signaling.

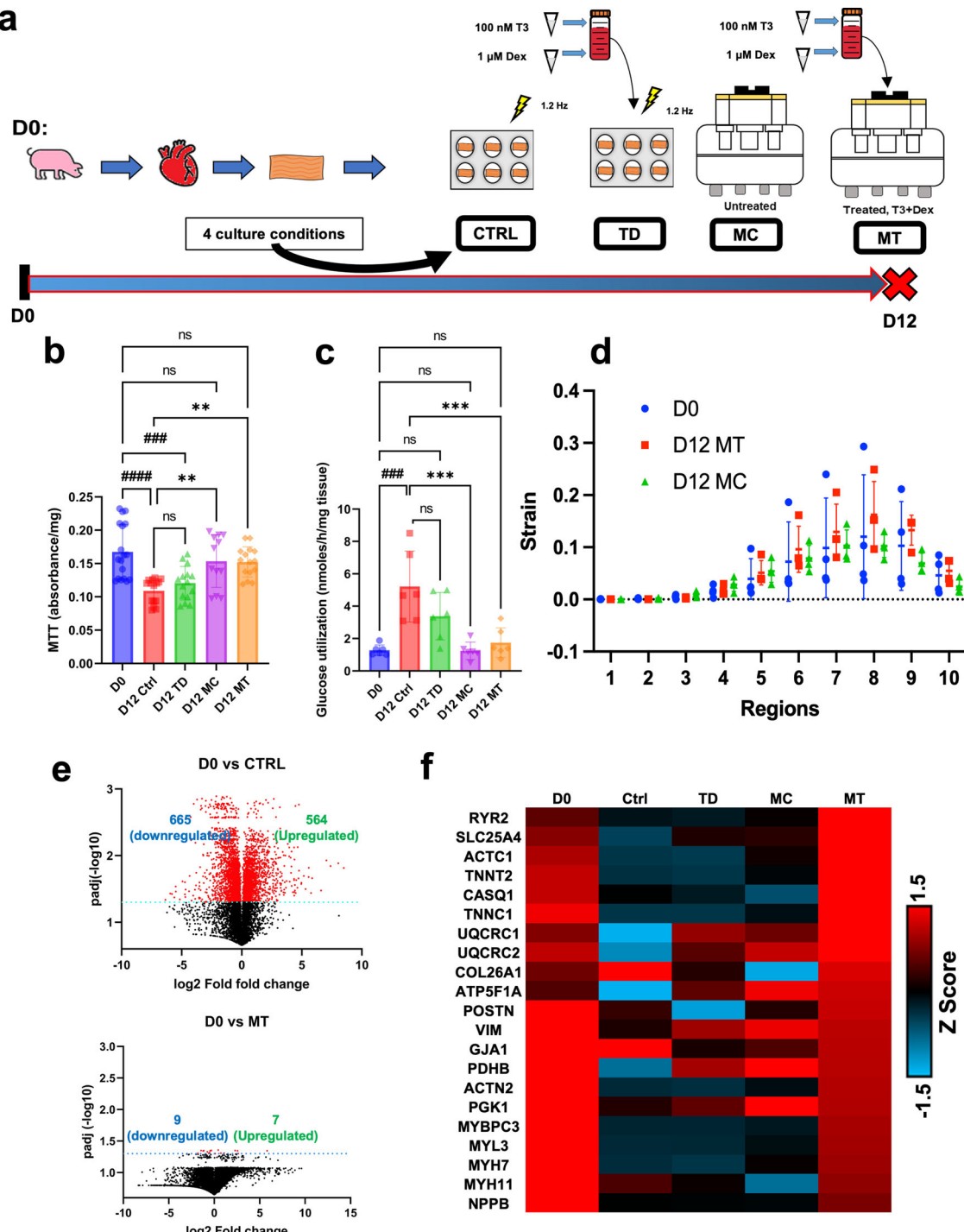

**Fig. 4 Combining mechanical and humoral stimulation (MT condition) preserve the heart slices for 12 days. a** Schematic representation of the experimental design depicting the four culture conditions that were used to assess the effect of combining mechanical stimulation and T3/Dex addition to the culture media over the course of 12 days. **b** Bar graph shows quantification of viability 12 days post culture in all 4 culture conditions (Ctrl, TD, MC, and MT) compared to fresh heart slices (D0) ($n = 18$ (D0), 15 (D12 Ctrl, D12 TD and D12 MT), 12 (D12 MC) slices/group from different pigs, one-way ANOVA test is performed; ####$p < 0.0001$, ###$p < 0.001$ compared to D0 and **$p < 0.01$ compared to D12 Ctrl). **c** Bar graph shows the quantification of glucose flux 12 days post culture in all 4 culture conditions (Ctrl, TD, MC, and MT) compared to fresh heart slices (D0) ($n = 6$ slices/group from different pigs, one-way ANOVA test is performed; ###$p < 0.001$, compared to D0 and ***$p < 0.001$ compared to D12 Ctrl). **d** Strain analysis plot for fresh (blue), day 12 MC (green), and day 12 MT (red) tissue across the ten regional points of the tissue slice ($n = 4$ slices/group, one-way ANOVA test is performed; no significant difference between the groups). **e** Volcano plots showing the differentially expressed genes in fresh heart slices (D0) compared to heart slices cultured under static conditions (Ctrl) or MT conditions (MT) for 10–12 days. **f** Heatmap for the sarcomeric genes for heart slices cultured under each culture condition. Error bars are representative of the Mean ± SD.

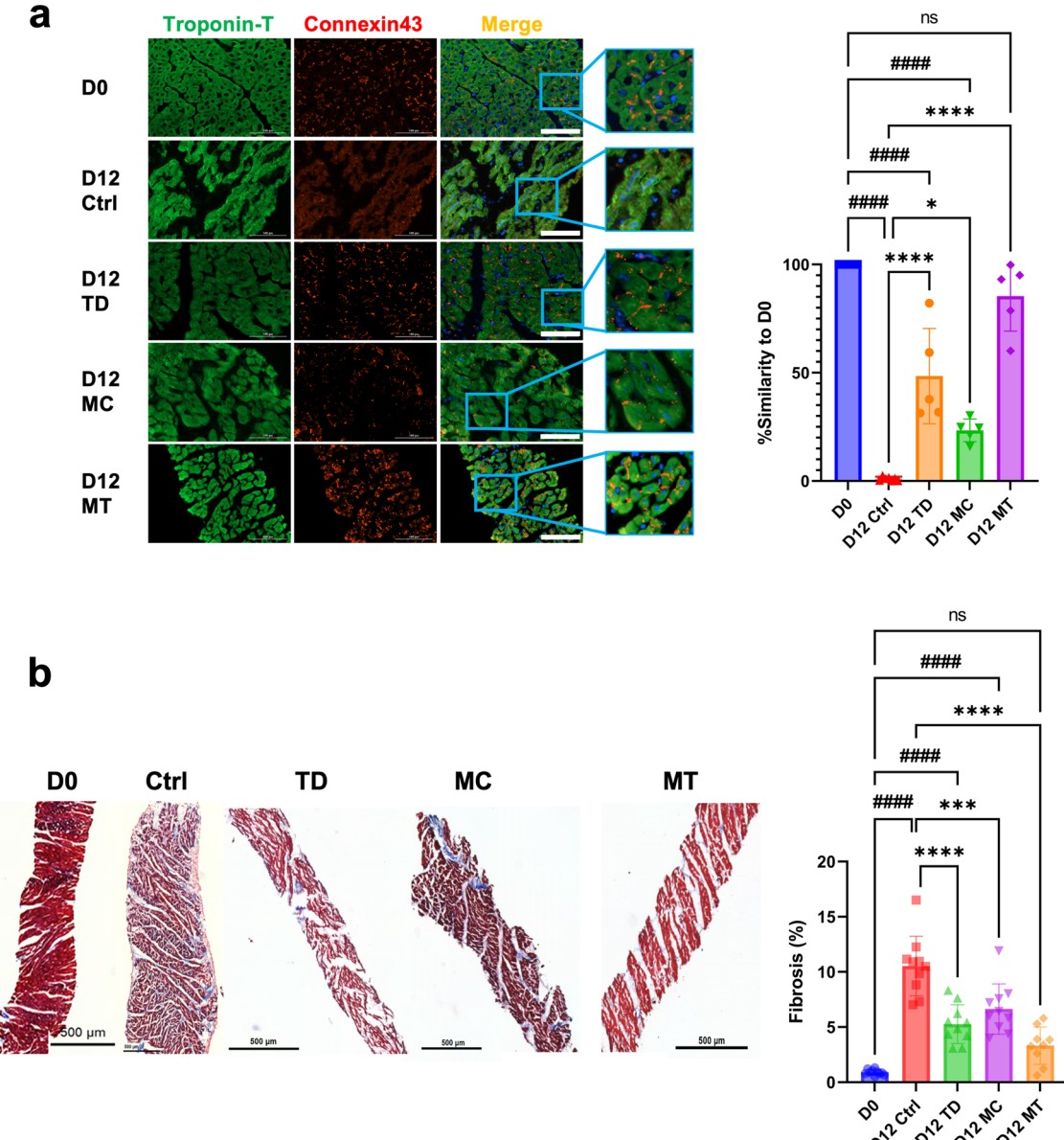

**Fig. 5 Cardiac tissue structure in slices cultured for 12 days under MT conditions is similar to fresh heart slices. a** Representative immunofluorescence images for troponin-T (green), connexin 43 (red), and DAPI (blue) for freshly isolated heart slices (D0) or heart slices cultured for 12 days under all four culture conditions (Scale bar $= 100\,\mu m$). Artificial intelligence quantification of the heart tissue structural integrity ($n = 7$ (D0 and D12 Ctrl), 5 (D12 TD, D12 MC and D12 MT) slices/group from different pigs, one-way ANOVA test is performed; $####p < 0.0001$ compared to D0 and $*p < 0.05$, or $****p < 0.0001$ compared to D12 Ctrl). **b** Representative images and quantification for heart slices stained with Masson's trichrome stain (Scale bar $= 500\,\mu m$) ($n = 10$ (D0, D12 Ctrl, D12 TD, and D12 MC), 9 (D12 MT) slices/group from different pigs, one-way ANOVA test is performed; $####p < 0.0001$ compared to D0 and $***p < 0.001$, or $****p < 0.0001$ compared to D12 Ctrl). Error bars are representative of the Mean ± SD.

## Discussion

Translational cardiovascular research requires cellular models capable of faithfully replicating the cardiac milieu. In this study, a CTCM device that can stimulate ultra-thin heart slices was developed and characterized. The CTCM system encompasses physiological synchronized electromechanical stimulation coupled with humoral enrichment with T3 and Dex. By subjecting pig heart slices to these factors, the viability, structural integrity, metabolic activity, and transcriptional expression were all maintained similar to fresh heart tissue for 12 days in culture. Furthermore, overstretching the heart tissue incites overstretching-induced cardiac hypertrophy. Overall, these results confirm the critical role of physiological culture conditions in maintaining a normal cardiac phenotype and provide a platform for drug screening.

Multiple factors contribute to the optimal environment for cardiomyocyte function and survival. The most obvious of these factors are related to (1) cell-to-cell interactions, (2) electro-mechanical stimulation, (3) humoral factors, and (4) metabolic substrates. Physiological cell-to-cell interaction requires the presence of intricate three-dimensional multi-cell-type networks supported by an extracellular matrix. This complex interaction of cells is difficult to recreate in vitro through the co-culture of individual cell types but can be easily accomplished by utilizing the organotypic nature of the heart slices.

Mechanical stretching and electrical stimulation of cardiomyocytes are essential for preserving the cardiac phenotype[33–35]. While mechanical stimulation has been used extensively in hiPSC-CM conditioning and maturation, some elegant studies

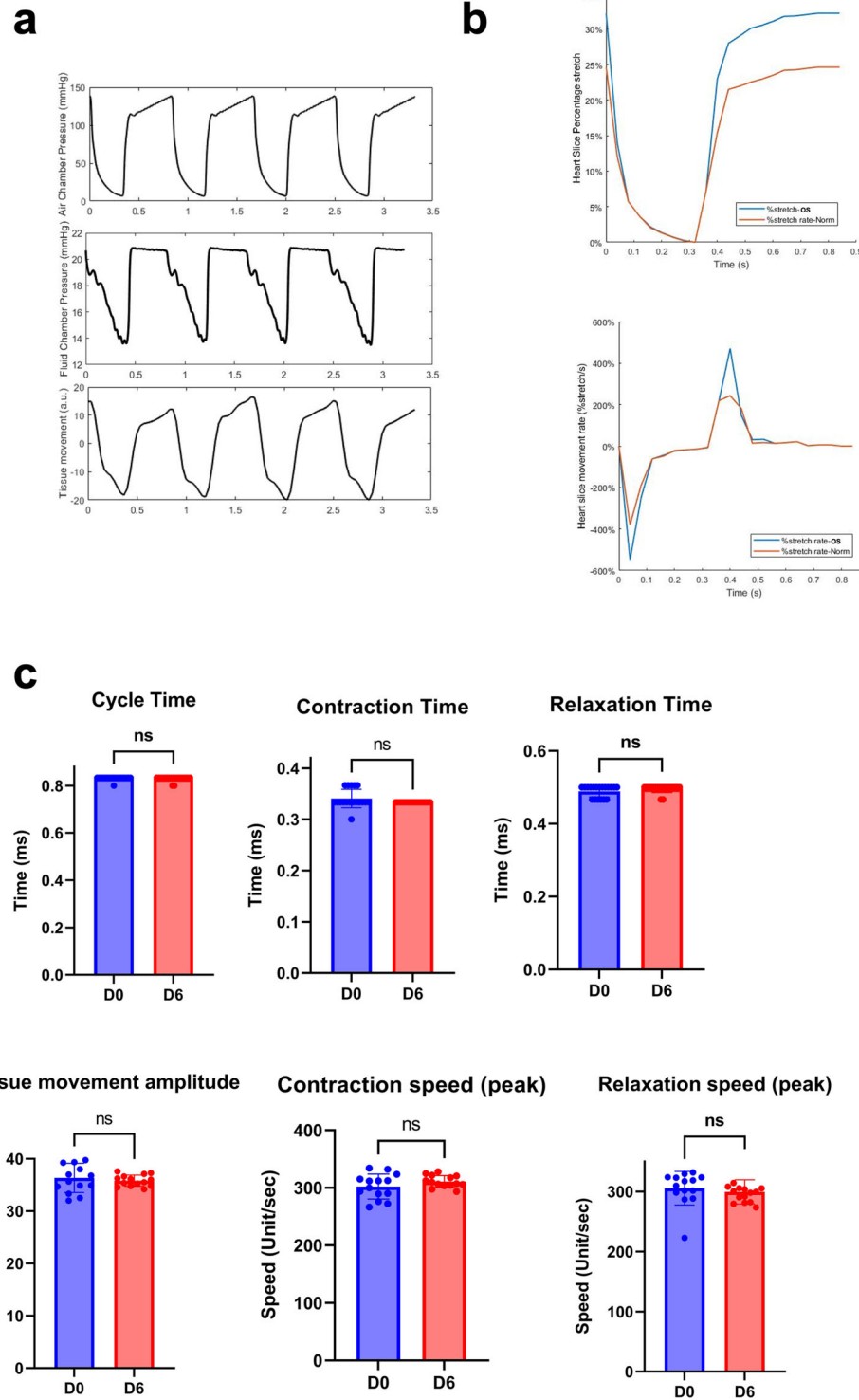

**Fig. 6 Induction of pathological overstretching in heart slices cultured in CTCM device. a** Representative traces of the air chamber pressure, fluid chamber pressure, and tissue movement measurements verified that the air chamber pressure changes the fluid chamber pressure, which induces a corresponding tissue slice movement. **b** Representative traces of the percent stretch and stretch rate of normal stretch (orange) and overstretched (blue) tissue slices. **c** Bar graphs showing quantification of the cycle time ($n = 19$ slices/group from different pigs), contraction time ($n = 18$–$19$ slices/group from different pigs), relaxation time ($n = 19$ slices/group from different pigs), tissue movement amplitude ($n = 14$ slices/group from different pigs), peak contraction speed ($n = 14$ slices/group from different pigs), and peak relaxation speeds ($n = 14$ (D0), 15 (D6) slices/group from different pigs), Two-tailed Student $t$-tests revealed no significant difference in any of the parameters, demonstrating that these parameters remained consistent over 6 days of overstretching culture. Error bars are representative of the Mean ± SD.

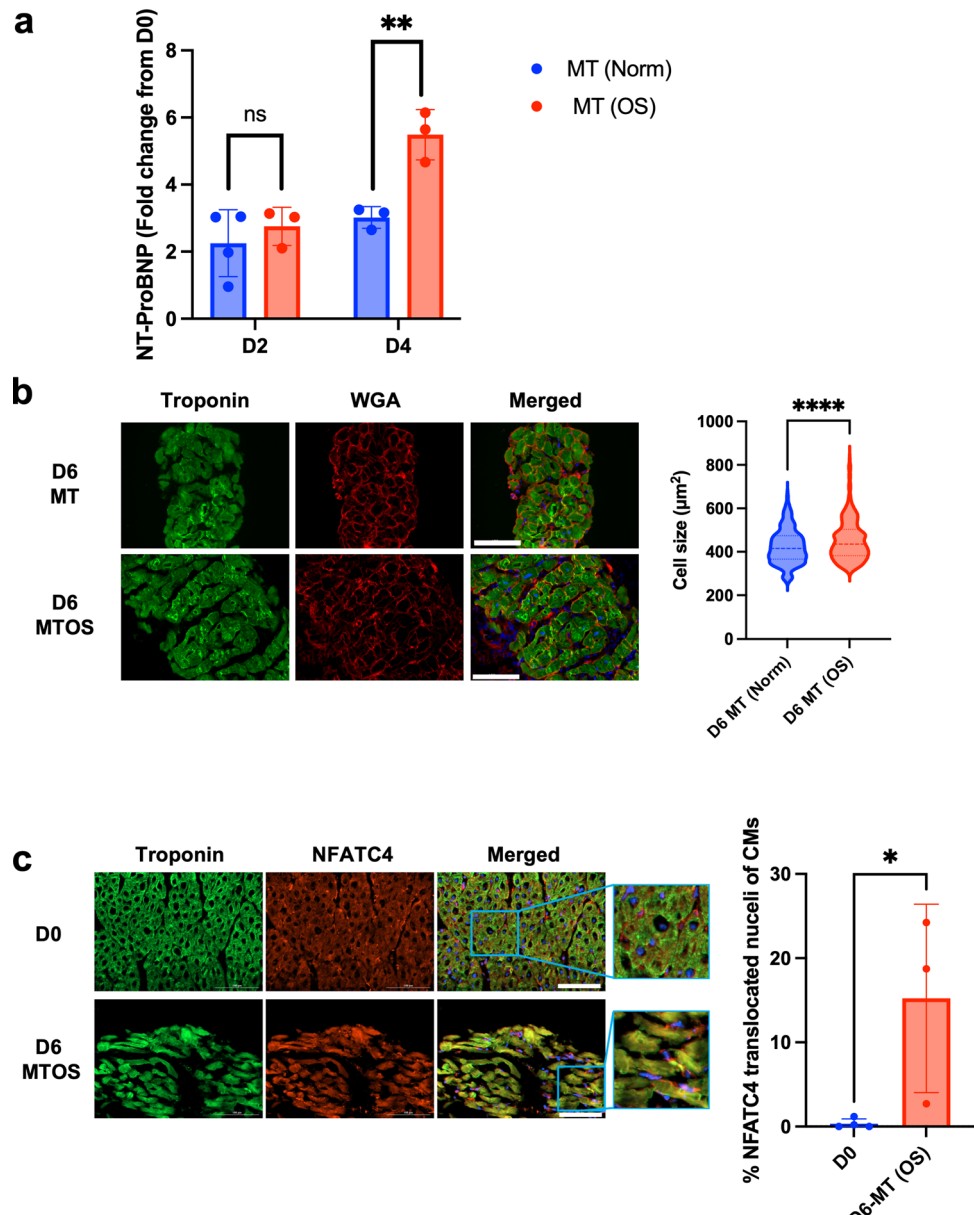

**Fig. 7 Overstretching in CTCM induces pathological hypertrophy. a** Bar graph quantification of NT-ProBNP concentration in culture media from heart slices cultured under MT normal stretch (Norm) or overstretching (OS) conditions ($n = 4$ (D2 MTNorm), 3 (D2 MTOS, D4 MTNorm, and D4 MTOS) slices/group from different pigs, Two-way ANOVA is performed; **$p < 0.01$ compared to normal stretch). **b** Representative images for heart slices stained with troponin-T and WGA (left) and cell size quantification (right) ($n = 330$ (D6 MTOS), 369 (D6 MTNorm) cells/group from 10 different slices from different pigs, Two-tailed Student $t$-test is performed; ****$p < 0.0001$ compared to normal stretch). **c** Representative images for day 0 and day 6 MTOS heart slices immunolabeled for troponin-T and NFATC4 and quantification of the translocation of NFATC4 to the nuclei of CMs ($n = 4$ (D0), 3 (D6 MTOS) slices/group from different pigs, Two-tailed Student t-test is performed; *$p < 0.05$). Error bars are representative of the Mean ± SD.

have recently attempted mechanical stimulation of heart slices in culture using uniaxial loading[17–19]. These studies have shown the positive influence of the 2D uniaxial mechanical loading on the cardiac phenotype during culture. In these studies, heart slices were either loaded with isometric stretch forces[17], linear auxotonic loading[18], or recreate the cardiac cycle using force transducer feedback and a stretcher actuator[19]. However, these methods adopted uniaxial stretching of the tissue without optimizing the culture media conditions, which resulted in the downregulation of multiple cardiac genes or the overexpression of genes related to pathological stretch response. The CTCM described here provides a three-dimensional electromechanical stimulation that mimics the native cardiac cycle in terms of cycle

timing and physiological stretches (25% stretch, 40% systole, 60% diastole, and 72 beats per minute). While this three-dimensional mechanical stimulation alone was not enough to maintain the tissue integrity, a combination of humoral stimulation using T3/Dex with mechanical stimulation was required to fully maintain the tissue viability, functionality, and integrity.

Humoral factors play an essential role in regulating adult cardiac phenotype. This has been emphasized in hiPS-CM studies that incorporate T3 and Dex in the culture medium to promote the maturation of the cells. T3 can influence amino acids, sugars, and calcium transport across the cell membrane[36]. In addition, T3 promotes the expression of MHC-α and downregulation of MHC-β, promoting fast-twitch myofibrils seen in mature

cardiomyocytes vs. slow-twitch myofibrils seen in fetal CMs[37]. The lack of T3 in hypothyroid patients can lead to the loss of myofibrillar striation and a reduced rate of tension development[37]. Dex acts on the glucocorticoid receptor and has been shown to increase cardiac contractility in ex vivo perfused hearts;[38] an improvement is thought to be related to effects on store-operated calcium entry (SOCE)[39,40]. In addition, Dex binds to its receptor; it induces a wide range of intracellular responses that suppress immune function and inflammation[30].

Our results demonstrate that physiological mechanical stimulation (MC) improved overall culture performance compared to Ctrl but cannot maintain viability, structural integrity, and cardiac expression for 12 days in culture. Incorporating T3 and Dex treatment in the CTCM culture (MT) resulted in improved viability compared to Ctrl and maintained transcriptional profile, structural integrity, and metabolic activity similar to fresh heart tissue for 12 days. Furthermore, a model of overstretch-induced cardiac hypertrophy modeling using the CTCM was achieved by controlling the extent of the tissue stretch, illustrating the versatility of the CTCM system. It should be noted that even though cardiac remodeling and fibrosis typically involve an intact organ with contributions from circulating cells that can provide relevant cytokines as well as phagocytosis and other remodeling factors, the heart slices can still emulate the fibrotic process in response to stress and injury by differentiating the resident fibroblast into myofibroblasts. This was previously assessed in this heart slice model[15]. It should be noted that the CTCM parameters can be modulated to simulate many conditions such as tachycardia, bradycardia, and mechanical circulatory support (mechanically unloaded heart) by changing the pressure/electrical amplitudes and frequencies. This allows for the system to be a medium-throughput platform for drug testing. The ability of the CTCM to model overstretch-induced cardiac hypertrophy opens the avenue for this system to be utilized in personalized therapy testing. In conclusion, the present study demonstrates that mechanical stretches and humoral stimulation are essential to maintain cardiac tissue slices in culture.

While the data shown here indicate that the CTCM is a very promising platform for modeling the intact myocardium, there are a few limitations to this culture method. The major limitation of the CTCM culture is that it applies continuous dynamic mechanical loading to the slices, which precludes the ability to actively monitor the heart slice contraction during each cycle. In addition, due to the small size of the heart slices (7 mm), there is a limited ability to perform contractile function assessment outside the culture system using traditional force transducers. In the current manuscript, we partially overcome this limitation through the assessment of the optical strain as a readout for the contractile function; however, this limitation will need further work and could be addressed in the future by implementing methods to optically monitor the functionality of the heart slices within the culture, such as optical mapping using calcium and voltage-sensitive dyes. Another limitation of the CTCM is that the working model is not manipulating the physiological pressures (preload and afterload). In the CTCM, pressure is induced in the opposite directions to reproduce the physiological stretches during diastole (full-length stretch) and systole (contracted length during electrical stimulation) in an oversized tissue by 25%. This limitation should be addressed in the future CTCM design by fully pressurizing the heart tissue from both sides and applying the accurate pressure-volume relationship that occurs within the heart chambers.

The overstretch-induced remodeling reported in this manuscript is limited in emulating only overstretch hypertrophic signaling. Therefore, this model could help study stretch-induced hypertrophic signaling[41,42] without humoral or neuronal factors

(which are lacking in this system). Further studies are needed to add multiplexity to the CTCM, such as co-culture with immune cells, circulating humoral plasma factors, and innervation with neuronal cell co-culture would advance the disease modeling capacity of the CTCM.

## Methods

**Heart tissue collection from pigs.** Thirteen pigs were used in the current study. All animal procedures were in accordance with the institutional guidelines and approved by the University of Louisville Institutional Animal Care and Use Committee. The aortic arch was clamped, and hearts were perfused with 1 L of sterile cardioplegia solution (110 mM NaCl, 1.2 mM CaCl₂, 16 mM KCl, 16 mM MgCl₂, 10 mM NaHCO₃, 5 units/ml heparin, pH to 7.4); the hearts were preserved in ice-cold cardioplegic solution until transported to the lab on ice which is usually <10 min.

**CTCM device fabrication.** The CTCM devices were designed in SolidWorks Computer-Aided Design (CAD) software. The culture chamber, separating plate, and air chamber were manufactured using CNC (computer numerical control) machining out of clear acrylic plastic. The 7 mm support rings were center-lathed from high-density poly-ethylene plastic (HDPE) and fitted with an O-ring groove to accommodate silicone O-rings for sealing the media below. A thin silicone membrane separates the culture chamber from the separating plate. The silicone membrane was laser-cut out of a 0.02" silicone sheet, durometer 35 A. The bottom and top silicone gaskets were laser-cut from a 1/16" silicone sheet, durometer 50 A. 316 L stainless steel screws and wing nuts were used to hold the device together to create an air-tight seal.

A custom printed circuit board (PCB) was designed to integrate with the C-PACE-EM system. Swiss machine headers outlets on the PCB are connected to a graphite electrode via silver-plated copper wires and 0-60 bronze screws threaded into the electrodes. The PCB fits into a three dimensionally printed device cover.

The CTCM device is controlled by a programmable pneumatic driver (PPD) that can induce a controlled cyclic pressure similar to the cardiac cycle. As the pressure within the air chamber increases, the flexible silicone membrane will distend upward, displacing the culture medium under the tissue slices. The tissue slices will then be stretched by this fluid displacement, mimicking physiological cardiac stretch during diastole. At the peak of this diastole, electrical stimulation is applied through the graphite electrodes, and the air chamber pressure is decreased, allowing the tissue slices to contract. Within the airline, there is a hemostatic valve with a pressure probe sensor that detects the pressure of the air system. The pressure detected by the pressure sensor is fed into a data acquisition device connected to a laptop. This allows for the continuous monitoring of the pressures within the air chamber. When the maximum air chamber pressure (80 mmHg for norm and 140 mmHg for OS) is reached, the data acquisition device is told to send a signal to the C-PACE-EM system inducing a 2 ms biphasic electrical voltage signal set at 4 V.

**Heart slice culture.** Heart slices were obtained, and 6-well culture conditions were performed as described below: the collected heart is transferred from the transfer container to a tray containing cold (4 °C) Cardioplegia solution. The left ventricle is isolated using sterile blades and cut into 1–2 cm³ blocks. These tissue blocks are attached to tissue supporters using tissue glue and placed into the tissue bath of a vibrating microtome containing Tyrode's solution with continuous oxygenation (3 g/L 2,3-butanedione monoxime (BDM), 140 mM NaCl (8.18 g), 6 mM KCl (0.447 g), 10 mM D-glucose (1.86 g), 10 mM HEPES (2.38 g), 1 mM MgCl₂ (1 mL of 1 M solution), 1.8 mM CaCl₂ (1.8 mL of 1 M solution), up to 1 L of ddH₂O). The vibrating microtome is set to cut 300 µm slices at a frequency of 80 Hz, horizontal vibration amplitude of 2 mm, and advance speed of 0.03 mm/s. The tissue bath is surrounded by ice to maintain a chilled solution and keep the temperature at 4 °C. Tissue slices are transferred from a microtome bath to a holding bath containing continuously oxygenated Tyrode's solution on ice until enough slices are obtained for one culture plate. For transwell culture conditions, tissue slices are adhered to sterilized polyurethane 6 mm wide supports and placed into six-well culture plates containing 6 mL of optimized culture media (Medium 199, 1x ITS supplement, 10% FBS, 5 ng/mL VEGF, 10 ng/mL FGF-basic, and 2X Antibiotic-Antimycotic). Electrical stimulation (10 V, paced at 1.2 Hz) was applied to the tissue slices through C-Pace lids. For TD conditions, fresh T3 and Dex were added at 100 nM and 1 µM concentrations at each media change. The culture media is oxygenated before every media change three times a day. Tissue slices were cultured in an incubator set to 37 °C with 5% CO₂.

For CTCM culture, the tissue slices were positioned on a custom three–dimensional-printed apparatus within a petri dish containing modified Tyrode's solution. The device is designed to oversize the heart slice by 25% of the support ring area. This is done to ensure that the heart slice is not over-stretched once it has been transferred from the Tyrode's solution to culture media and during the diastolic phase. Using histoacryl glue, the 300 µm thick slices were fixed to 7-mm diameter support rings. Once the tissue slice was attached to the support ring, the excess tissue slice was trimmed off, and the attached tissue slice was placed

back into the Tyrode's solution bath on ice (4 °C) until enough slices were prepared for one device. The processing time should not exceed 2 h in total for all devices. Once 6 tissue slices were attached to their support rings, the CTCM device was assembled. The CTCM culture chamber was prefilled with 21 mL of pre-oxygenated culture media. The tissue slices were transferred into the culture chamber, and all air bubbles were carefully removed with a pipette. Then the tissue slice was guided to a well and gently pressed into place. Lastly, the electrode cover was placed onto the device, and the device was transferred to an incubator. The CTCM was then connected to the air tubes and the C-PACE-EM system. The pneumatic driver was switched on and the airflow valve opened to the CTCM device. The C-PACE-EM system was set to provide 4 V at 1.2 Hz at 2 ms biphasic stimulation. The culture medium was changed twice daily, and the electrodes were replaced once a day to avoid graphite buildup from the electrodes. If necessary, the tissue slices can be removed from their culture wells to displace any air bubbles that may be trapped below. For MT conditions, the T3/Dex treatment was added fresh with every media change at 100 nM T3 and 1 μM Dex. The CTCM devices were cultured in an incubator set to 37 °C with 5% $CO_2$.

**Heart slice mechanical stretches assessment using MUSCLEMOTION.** A custom camera system was developed to obtain a stretch trace of the heart slices. A DSLR camera (Canon Rebel T7i, Canon, Tokyo, Japan) was used with a Navitar Zoom 7000 18-108 mm Macro Lens (Navitar, San Francisco, CA). After changing the media with a fresh culture medium, the imaging was conducted at room temperature. The camera was placed at a 51° angle, and videos were recorded at 30 fps. Firstly, an open-source software (MUSCLEMOTION[43]) was used with Image-J to quantify the movement of the heart slices. A mask was created using MATLAB (MathWorks, Natick, MA, USA) to define the region of interest of the beating heart slice to avoid any noise. The manually segmented mask was applied to all images in the frame sequence and then fed to the MUSCLEMOTION plugin. MUSCLE motion uses the average pixel intensity within each frame to quantify its movement relative to a reference frame. The data was recorded, filtered, and used to quantify the cycle time and assess the tissue stretches during the cardiac cycle. Post-processing of recorded videos was performed using a zero-phase first-order digital filter. To quantify tissue stretch (peak to peak), peak analysis was performed to distinguish the peaks and valleys of the recorded signal. In addition, detrending was performed to remove signal drift using a 6th-order polynomial. Software code was developed in MATLAB to detect overall tissue movement, cycle time, relaxation time, and contraction time (Supplementary software code[44]).

**Strain analysis.** For strain analysis, using the same videos generated for the mechanical stretch assessment, we first tracked the two images that represent the peaks of the movements (highest point of movement (Up) and lowest point (Down)) according to the MUSCLEMOTION software. Then we segmented the tissue areas and applied the shape from shading algorithm[45] on the segmented tissues (Supplementary Fig. 2a). The segmented tissue is then divided into ten sub-surfaces and the strain for each surface is calculated using the following equation: Strain $= (S_{up} - S_{down})/S_{down}$ where $S_{up}$ and $S_{down}$ are distances of the shape from shading of the up and down tissues, respectively (Supplementary Fig. 2b).

**Heart slice fixation and mounting.** Heart slices were fixed in 4% paraformaldehyde for 48 h. Fixed tissue underwent dehydration in 10% then 20% sucrose for 1 h, followed by 30% sucrose overnight. The slices were then embedded in an optimal cutting temperature compound (OCT compound) and gradually frozen in an isopentane/dry ice bath. OCT embedded blocks were stored at −80 °C until sectioning. Slides were prepared in 8 μm thick sections.

**Immunostaining.** To remove the OCT from the heart slices, the slides were heated on a heat block for 5 min at 95 °C. 1 mL of PBS was added to each slide and incubated at room temperature for 30 min Sections were then permeabilized by setting for 15 min with 0.1% Triton-X in PBS at room temperature. To prevent non-specific antibody binding to the samples, 1 mL of 3% BSA solution was added to the slides and incubated for 1 h at room temperature. The BSA was then discarded, and the slides were washed with PBS. Using a wax pen, each sample was marked off. The primary antibodies (1:200 dilution in 1% BSA) (Connexin 43 (Abcam; #AB11370), NFATC4 (Abcam; #AB99431) and Troponin-T (Thermo Scientific; #MA5-12960), were added to the section for 90 min, followed by the secondary antibodies (1:200 dilution in 1% BSA) Anti-mouse Alexa Fluor 488 (Thermo Scientific; #A16079), Anti-rabbit Alexa Fluor 594 (Thermo Scientific; #T6391) for another 90 min separated by three washes with PBS. To distinguish the target staining from the background, we only used a secondary antibody as a control. Finally, the DAPI nuclear stain was added, and the slides were mounted in vectashield (Vector Laboratories) and sealed with nail polish. Immunofluorescence imaging was performed using a Cytation 1 high content imager (20X lens magnification) and a Keyence microscope using 40X lens magnification.

For WGA staining, WGA-Alexa Fluor 555 (Thermo Scientific; #W32464) was used at 5 μg/mL in PBS and was applied to fixed sections for 30 min at room temperature. The slides were then washed with PBS, and Sudan Black was added to each slide and incubated for 30 min. The slides were then washed with PBS and

vectashield mounting media was added. Slides were imaged on a Keyence microscope using 40X lens magnification.

**Masson's Trichrome stain.** OCT was removed from the samples as described above. Once the OCT was removed, the slides were submerged in Bouin's solution overnight. The slides were then rinsed with distilled water for 1 h, then placed in Biebrich Scarlet-acid Fuchsin solution for 10 min. The slides were then washed with distilled water and placed in 5% phosphomolybdic/5% phosphotungstic acid solution for 10 min. Without rinsing, the slides were directly transferred into an aniline blue solution for 15 min. Then the slides were rinsed with distilled water and placed in a 1% acetic acid solution for 2 min. The slides were dried in 200 proof ethyl alcohol and transferred into xylene. Stained slides were imaged using a Keyence microscope with 10× lens magnification. The percentage area of fibrosis was quantified using the Keyence Analyzer software.

**MTT assay.** CyQUANT™ MTT Cell Viability Assay (Invitrogen, Carlsbad, CA), catalog # V13154, was used according to the manufacturer's protocol with some modifications. In details, A 6 mm surgical punch was used to ensure similar tissue size when performing the MTT assay. The tissues were each placed in a well of 12 well plates containing MTT substrate according to the manufacturer's protocol. The slices were incubated for 3 h at 37 °C and viable tissue metabolized the MTT substrate creating a purple formazan compound. The MTT solution was replaced with 1 ml of DMSO and incubated at 37 °C for 15 min to extract the purple formazan from the heart slices. Samples were diluted to 1:10 in DMSO within a clear bottom 96 well plate, and the intensity of the purple color was measured using a Cytation plate reader (BioTek) at 570 nm. The readings were normalized to the weight of each heart slice.

**Tritiated glucose utilization assay.** The media of heart slices was changed with a culture medium containing 1 μCi/ml [5-3H]-glucose (Moravek Biochemicals, Brea, CA, USA) for glucose utilization assay, as described previously[46,47]. Following incubation for 4 h, 100 μl of media was added to a cap-less microcentrifuge tube containing 100 μl of 0.2 N HCl. Then the tube was placed in a scintillation vial containing 500 μl of dH₂O to allow for evaporation diffusion of [³H]₂O at 37 °C for 72 h. Then, the microcentrifuge tubes were removed from the scintillation vials, and 10 ml of scintillation fluid was added. Scintillation counting was performed using a Tri-Carb 2900TR Liquid Scintillation Analyzer (Packard Bioscience Company, Meriden, CT, USA). Glucose utilization was then calculated, with considerations for the specific activity of [5-³H]-glucose, incomplete equilibration and background, dilution of [5-³H]-to unlabeled-glucose, and scintillation counter efficiency. Data were normalized to the weight of the heart slices.

**RNAseq.** RNA was isolated from the heart slices by using the Qiagen miRNeasy Micro Kit, #210874, following the manufacturer's protocol after homogenizing tissue in Trizol. RNAseq library preparation, sequencing, and data analysis were performed as described below:

1 μg RNA per sample was used as input material for the RNA library preparations. Sequencing libraries were generated using NEBNext UltraTM RNA Library Prep Kit for Illumina (NEB, USA) following the manufacturer's recommendations and index codes were added to attribute sequences to each sample. Briefly, mRNA was purified from total RNA using poly-T oligo-attached magnetic beads. Fragmentation was carried out using divalent cations under elevated temperature in NEBNext First Strand Synthesis Reaction Buffer (5X). First strand cDNA was synthesized using random hexamer primer and M-MuLV Reverse Transcriptase (RNase H-). Second strand cDNA synthesis was subsequently performed using DNA Polymerase I and RNase H. Remaining overhangs were converted into blunt ends via exonuclease/polymerase activities. After adenylation of 3′ ends of DNA fragments, NEBNext Adaptor with hairpin loop structure were ligated to prepare for hybridization. In order to select cDNA fragments of preferentially 150–200 bp in length, the library fragments were purified with AMPure XP system (Beckman Coulter, Beverly, USA). Then 3 μl USER Enzyme (NEB, USA) was used with size-selected, adaptorligated cDNA at 37 °C for 15 min followed by 5 min at 95 °C before PCR. Then PCR was performed with Phusion High-Fidelity DNA polymerase, Universal PCR primers and Index (X) Primer. At last, PCR products were purified (AMPure XP system) and library quality was assessed on the Agilent Bioanalyzer 2100 system. Then the cDNA libraries are sequenced using Novaseq sequencer. Original image data file from Illumina is transformed to raw reads by CASAVA base recognition (Base Calling). Raw data are stored in FASTQ(fq) format files, which contain sequences of reads and corresponding base quality. HISAT2 is selected to map the filtered sequenced reads to the reference genome Sscrofa11.1. In general, HISAT2 supports genomes of any size, including those larger than 4 billion bases and most of the parameters are set to default. Spliced reads of RNA Seq data can be effectively aligned using HISAT2, which is the fastest system currently available, with equal or better accuracy than any other method.

The abundance of transcript reflects gene expression level directly. Gene expression level is estimated by the abundance of transcripts (count of sequencing) that mapped to genome or exon. Read counts is proportional to gene expression level, gene length and sequencing depth. FPKM (Fragments Per Kilobase of

transcript sequence per Millions base pairs sequenced) were calculated and differential expression P-values were determined using DESeq2 package. Then, We calculated the false discovery rates (FDRs) for each P-value with the Benjamini-Hochberg method 9 based on the built-in R function "p.adjust".

**qRT-PCR.** The RNA extracted from heart slices, was converted to cDNA at 200 ng/µl concentration using Thermo's SuperScript IV Vilo Master mix (Thermo, Cat # 11756050). qRT-PCR was run using Applied Biosystems microamp Endura plate optical 384-well clear reaction plate (Thermo, Cat # 4483319) with microamp optical adhesive film (Thermo, Cat # 4311971). The reaction mixture is consisted of 5 µl Taqman Fast Advanced Master mix (Thermo, Cat # 4444557), 0.5 µl Taqman Primer, and 3.5 µL $H_2O$ per well were mixed. The standard qPCR cycle was run and CT values were measured using an Applied Biosystems Quantstudio 5 Real-Time PCR instrument (384-well block; Product #A28135). Taqman primers were purchased from Thermo (GAPDH (Ss03375629_u1), PARP12 (Ss06908795_m1), PKDCC (Ss06903874_m1), CYGB (Ss06900188_m1), RGL1 (Ss06868890_m1), ACTN1 (Ss01009508_mH), GATA4 (Ss03383805_u1), GJA1 (Ss03374839_u1), COL1A2 (Ss03375009_u1), COL3A1 (Ss04323794_m1), ACTA2(Ss04245588_m1). All sample CT values were normalized to the housekeeping gene GAPDH.

**NT– ProBNP.** NT–ProBNP release in the culture media was assessed using an NT-ProBNP kit (pigs) (Cat # MBS2086979, MyBioSource) according to the manu-facturer protocol. Briefly, 250 µL of each sample and standard were added to each well in duplicate. Immediately following the addition of the samples, 50 µL of Detection Reagent A was added per well. The plate was shaken gently and sealed with a Plate Sealer. The plate was then incubated for 1 h at 37 °C. Then, the solution was aspirated, and the wells were washed with 350 µL of 1X Wash Solution 4 times, letting the Wash Solution incubate for 1–2 min each time. Next, 100 µL of Detection Reagent B was added per cell and sealed with a Plate sealer. The plate was gently shaken and incubated at 37 °C for 30 min. The solution was aspirated, and the wells were washed with 350 µL of 1X Wash Solution 5 times. 90 µL of Substrate Solution was added to each well, and the plate was sealed. The plate was incubated for 10-20 min at 37 °C. 50 µL of Stop Solution was added per well. The plate was immediately measured using a Cytation plate reader (BioTek) set at 450 nm.

**Statistics and reproducibility.** Power analyses were performed to choose the group sizes which will provide >80% power to detect a 10% absolute change in the parameter with a 5% Type I error rate. Tissue slices were randomized choices before experiments. All analyses were blinded with regard to conditions, and samples were decoded only after all data were analyzed. GraphPad Prism software (San Diego, CA) was used to perform all statistical analyses. For all statistics, $p$-values were considered significant at values <0.05. Two-tailed Student $t$-tests were performed for data with only 2 group comparisons. One-way or two-way ANOVA was used to determine the significance between multiple groups. The Tukey correction was applied to account for multiple comparisons when per-forming post hoc tests. RNAseq data had special statistical consideration to cal-culate the FDR and the p.adjust as summarized under the methods section.

**Reporting summary.** Further information on research design is available in the Nature Research Reporting Summary linked to this article.

## Data availability
The authors declare that all data supporting the findings of this study are available within the paper and its supplementary information files. Please see supplementary data 1 excel file for the RNAseq data and the data points for each figure. In addition, the RNAseq raw data are deposited in GEO repository under accession number GSE211110.

## Code availability
The authors declare that all data supporting the findings of this study are available within the paper and its supplementary information files. Supplementary software code file shows the computer codes used in analyses. Software code was deposited in Zenodo[44]: https://doi.org/10.5281/zenodo.7023518.

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

## Acknowledgements
The authors would like to thank the funding agencies that made this work possible: T.M.A.M. is supported by NIH grants R01HL147921 and P30GM127607 and American Heart Association grant 16SDG29950012. The authors also acknowledge NIH grants F32HL149140 (R.R.E.A.), P30GM127607 (B.G.H.), R01HL130174 (B.G.H.), R01HL147844 (B.G.H.), R01ES028268 (B.G.H.), P01HL78825 (R.B., B.G.H.), UM1HL113530 (R.B.), and 2U54HL120163. We also acknowledge the United States of America Department of Defense for the grant W81XWH-20-1-0419 (T.M.A.M.).

## Author contributions
J.M.M. and M.H.M.: experimental design, collection and analysis of molecular and cellular data, manuscript writing, and final approval of the manuscript; A.E.: Performed strain analysis; Q.O.: heart cutting, staining, and imaging; X-L.T., R.B.: Pig heart collection and manuscript writing, R.R.E.A. and B.G.H.: Metabolic analysis; A.S., C.L., and A.G.: immunostaining and trichrome staining and analysis; H.A., F.K., N.A., and A.S.E.: Artificial intelligence analysis for heart slice tissue integrity; G.G.: Engineering design and supervision of the CTCM and mechanical stimulation; T.M.A.M.: conception and design of the overall work, and provided funding. All authors have contributed to the manuscript writing and final approval of the manuscript.

## Competing interests
T.M.A.M. holds equities at Tenaya Therapeutics. G.G. consults for NuPulseCV. N.A. is the chief scientific officer and holds equities at Anabios. All other authors declare no competing interest.
