## [Peer Review File · Communications Biology]

Reviewers' comments:

Reviewer #1 (Remarks to the Author):

This paper describes a culture methodology for heart slices. The authors present an interesting loading system based on the variations of pressure between two chambers separated by the heart tissue. The tissue is field stimulated and the mechanical stimulation reproduces the timing of the in-vivo systolic and diastolic intervals. With this method, that can be applied to 6 slices at the same time, the authors are able to maintain the slices for longer, and when combined with Dexamethasone and T3, to improve their viability and structural similarity compared with fresh tissue. Most strikingly, they observe an almost identical transcriptomic profile compared with D0 culture. This paper is topical as it shows myocardial slices to be an interesting model for cardiovascular research. However, I have some important concerns:

Novelty: the idea is not novel. The authors play down strongly other work in the field. They claim erroneously that previous studies are limited to 24 hour culture and have a lower throughput. The paper by Fischer et al (Nat Comms 2019) shows a method that allows culture of human slices for up to 4 months and it is clearly superior to the current model. Several slices can be cultured at the same time and the development of force is clearly documented. Watson (Nat Comms 2019) also shows culture of rabbit slices for 5 days without any loss in contractility. Pitoulis et al (Cardiovasc Res 2021) shows culture of rat slices up to 3 days. The latter paper also clearly documents the effect of overloading slices to obtain remodelling in vivo. This paper uses a much more sophisticated method and it is surprising that the authors mention this paper only superficially.

Novelty of mechanical stimulation: this concept has been exploited clearly in the Fischer and Watson paper and optimised in the Pitoulis paper where an accurate recapitulation of the in vivo force-length relationship of the working myocardium is incorporated in the model. The current paper only offers a rudimentary and superficial approximation.

The method of simulating mechanical load is not physiological: in the working myocardium, luminal pressures (afterload) increase during systole and decrease during diastole. In the method presented here, pressures are induced in the opposite direction. Can the authors supply a plot of chamber pressure and slice shortening (volume) and compare it to a physiological human pressure -volume loop of the LV?

Given the above and the observation that Des and T3 only partially prevent remodelling (supplementary Fig 2) it is surprising that the transcriptomic profile was completely unchanged in the MT group. This result is unexpected and requires validation, at least for selected genes, using rtPCR and for protein levels using Western blotting.

The strongest limitation of this study, recognized by the authors, is the absence of knowledge about contractility of the slices. It is deceiving to present data for slice movement implying that this is an active phenomenon while it seems to me that this is mostly passive, due to pressure changes in the chambers. How can the authors show that the slices contract at all upon stimulation, nevermind quantify this contraction? If the slices contract regularly, it is also possible that they contract in response to mechano-electrical feed back and not the electrical stimulation. It is imperative that this is documented.

Staining: I do not understand the pictures for muscle staining. There seem to be vacuoli or round negative structures that do not resemble the structure of the myocardium. In addition, the fiber orientation of the slices seem to be random, with myocytes oriented at different angles even in the same image (Figure 7). This makes quantification of cell size impossible.

Reviewer #2 (Remarks to the Author):

GENERAL COMMENTS

Miller and colleagues present a manuscript describing the generation of a culture system for pig heart slices and presumably human heart slices. This work advances on the prior studies from the Mohamed lab by now adding dynamic preload stretch to the slices as well as supplementing the media with tri-iodo-thyronine and dexamethasone. This results in the ability to maintain slices for up to 12 days in culture with preserved structural integrity, viability and gene expression. This is a significant advance over previous studies which showed preserved viability for a day or a few days. The authors also suggest that overstretching the tissues mimics cardiac hypertrophy. The manuscript has a number of strengths including describing the custom system to maintain the cardiac tissue culture model as well as the striking histological and gene expression results showing preservation of the tissue for 12 days. This represents an important advance for the field, but there are some limitations to the manuscript in its current form as follow:

- 1) Is the electrical stimulation capturing the tissue and triggering contraction? How do the authors determine this?
- 2) Is electrical stimulation necessary for the results? The simpler the system, the better, so could the stretching protocol be adequate to preserve structure and viability?
- 3) The authors acknowledge that they cannot determine the contraction of the constructs with the constant cycling pressure protocol, but couldn't some functional measurements be made by transiently turning off the pneumatic device and doing electrical stimulation alone? This could provide a contractile functional output.
- 4) The authors described differences in fibrosis (e.g. Fig. 3C) in the various slice interventions, but this is difficult to understand in an ex vivo slice preparation. Fibrosis typically involves an intact perfused organ with contributions from circulating cells that can provide relevant cytokines as well as phagocytosis and other remodeling factors. How do the authors envision this is possible in a 300 μm slice of tissue?
- 5) The authors claim of "pathological remodeling ... providing a proof of principle that the CTCM device can be used to induce overload cardiac hypertrophy." seems overstated. Is there really remodeling of the tissue or is this just an acute effect of stretch? It requires more than BNP and cell size measurements to demonstrate pathological remodeling. Furthermore, pressure overload models that generate cardiac hypertrophy typically result from an increase in afterload, not preload, as this model proposes to use. This caveat should be addressed.
- 6) Statistical analysis appears appropriate.

MINOR COMMENTS

1. Line 53, "Additionally, using of animal models to create pharmacokinetic profiles of drugs is relatively expensive..." I do not understand how this is relevant rationale for the model which does not address pharmacokinetics in any way.
2. Line 66, "the immature nature of the hiPS-CMs and loss of the multicellular complexity of the heart tissue..." There really is no loss but rather 'lack of ...'
3. Line 115, "The custom PCB..." Please define PCB.
4. Line 128, "When the maximum pressure is reached..." What defines the maximum pressure. It looks like this is a fixed value that is selected by investigators. Please clarify.
5. Figure 1B legend does not describe the schematic shown in Figure 1B
6. Line 620, "...representative trances..." should be "...representative traces..."
7. Figure 7A Y axis label, consider NT-Pro-BNP Fold of D0 to clearly define what is plotted.

Reviewer #3 (Remarks to the Author):

I have reviewed the manuscript submitted by Dr. Mohamed and colleagues, "Biomimetic Cardiac Tissue Culture Model (CTCM) to Emulate Cardiac Physiology and Pathophysiology ex vivo" (Manuscript Number: COMMSBIO-22-0379-T).

The study describe a method and a device to maintain in culture ventricular slices for prolonged periods of time (12 days) without losing the main characteristics of the native tissue. Mohamed and colleagues have determine in their study that electrical and mechanical stimulation, accompanied by a defined optimized culture media, allows to maintain not only viability, but also structural, molecular and metabolic integrity of the tissue for at least 12 days.

A dozen of papers have been published in the last few years by Dr Mohamed's group and others describing different approaches to maintain myocardial tissue in culture (from humans and other species), showing how to improve its similarity with native tissue, or extending the culture period. In this study, the authors take a step forward, and apply for the first time a dynamic mechanical load to a myocardial tissue from a big mammal (pig), and successfully maintain native characteristics for 12 days.

The development of myocardial tissue culture models, representative of the native physiology and pathophysiology of the heart, will be of great value in the future and immediate horizon of cardiovascular research. Consolidation of methods to achieve this is key to extending its use among a wide range of researchers. With this work, Dr. Mohamed's group has gone a step further in the establishment of such models.

Experimental design is appropriate, the manuscript is clearly written, and provides sufficient detail to allow its replication. I congratulate the authors for a hard work.

As outlined by the authors, a major weakness of the study is the lack of contractility studies or electrophysiological characterization (for example employing microelectrode arrays or optical mapping). Even when cellular structure and gene expression appear to be maintained, electrophysiological assessment should be done to decipher conduction alterations, or arrhythmogenic susceptibility. This should be evaluated in future studies.

I would like to point out some specific comments and suggest some minor corrections or additions. This could facilitate better understanding of the protocols and improve reproducibility.

- In the abstract section, in background, authors talk about their previous studies with human and pig hearts, but nowhere in the abstract is it indicated that the current work is done on pig tissue. Please outline that current study is performed using porcine myocardial tissue.

- In line 42, authors say that "this is the first proof of concept for the ability of the CTCM to emulate cardiac pathophysiology", but this has already been pointed out by Pitoulis et al, albeit with a lower throughput approach and lesser characterization, as authors outline in the discussion. So I would suggest lowering this statement.

- line 45, "that incorporates physiological mechanical and humoral cues", it would be better to say electromechanical instead of mechanical.

- Introduction section: lines 85-95. It would be helpful for the reader to outline in which species experiments have been done, as mice/rats hearts do not respond the same way as pig/human hearts to some procedures, like cardioplegic arrestment. For example: line 86, "to keep pig and human cardiac tissue slices", line 89 "to culture heart slices from different mammals for extended period of time". Line 91, outline this work is done in pigs (not in humans, but not in rodents).

Methods section:

- line 104: indicate approximate transport times, how long hearts have been maintained in cardioplegia solution?
- a novel reader not familiar with the previously published cardiac tissue culture experiments could appreciate a short description of the approach, not only the reference of the paper. A short description like "electrically stimulating the slices maintained with optimized culture medium".
- line 137: Tyrode's composition is not described in this paper. List the components of the modified Tyrode's solution employed, or cite a reference of a work with the same Tyrode's solution. Outline also the temperature at which is used.
- line 142: specify approximate time needed for processing and if Tyrode's is cold or at room temperature.
- line 145: is culture media pre-oxygenated as in previous works?
- line 157: it is not clear for me if these experiments are done in the incubator or out of it, at 37°C or at room temperature, with culture medium or with Tyrode's.

Results section:

- line 267: figure 1 b exists, but is not explained in figure 1 legend. Indeed, in legends, what is written as 1 B is in fact 1 C, and 1 C is 1D. References in the main text are correct, but you have to explain 1B in the legend, and move the numeration of 1 C and D accordingly.
- line 300, it would be useful for the reader to include here a brief description of what parameters are used to quantify structural integrity (not only the reference for the published work). I think it would be interesting also to show a graph with the quantification of troponin and connexin, additionally to the representative images showed and the graph of the % of similarity calculated with the AI framework.
- line 302, this is showed in figure 3b, but is not indicated. The same with "figure 3c" in line 304.
- lines 327-341. Insert the references for "figure 4a" (in line 331 approx.), "figure 4b" (line 332), and "figure 4c" (line 338).
- line 347, this is figure 4d instead of 5D. Line 352, this is figure 4e instead of 5E.
- about RNAseq and transcriptional study: I think it would be interesting to outline the 16 genes differentially expressed in the MT slices, for example in a small table in the supplement, or a graph with supplementary figure 5.
Figure 4e show sarcomeric genes, but this is not clearly indicated in figure 4e legend, where it states "major cardiac gene".
An additional comment regarding RNAseq study: taking into account that electrophysiology of the cultured slices has not been explored in this work, it could be interesting to show a heatmap, similar to the 4e figure, for the major genes of the ion channels involved in action potential. This could add evidence of electrophysiological maintenance in the MT slices, even when it should be further assessed in future optical mapping or microelectrode array studies.

Typographical errors:

- line 413: evinced -> evidenced
- line 620: trances -> traces
- line 636: full stop before Artificial intelligence

Response to Reviewers:

We would like to thank the editor and the reviewers for their careful consideration of our manuscript and their valuable comments and helpful suggestions, which have significantly improved the manuscript. Below is our point-by-point response to the reviewers' comments

Reviewers' comments:

Reviewer 1:

Comment 1: *This paper describes a culture methodology for heart slices. The authors present an interesting loading system based on the variations of pressure between two chambers separated by the heart tissue. The tissue is field stimulated and the mechanical stimulation reproduces the timing of the in-vivo systolic and diastolic intervals. With this method, that can be applied to 6 slices at the same time, the authors are able to maintain the slices for longer, and when combined with Dexamethasone and T3, to improve their viability and structural similarity compared with fresh tissue. Most strikingly, they observe an almost identical transcriptomic profile compared with D0 culture. This paper is topical as it shows myocardial slices to be an interesting model for cardiovascular research.*

Response: We would like to thank the constructive reviewer for his comment and his careful review of our manuscript.

Comment 2: *Novelty: the idea is not novel. The authors play down strongly other work in the field. They claim erroneously that previous studies are limited to 24 hour culture and have a lower throughput. The paper by Fischer et al (Nat Comms 2019) shows a method that allows culture of human slices for up to 4 months and it is clearly superior to the current model. Several slices can be cultured at the same time and the development of force is clearly documented. Watson (Nat Comms 2019) also shows culture of rabbit slices for 5 days without any loss in contractility. Pitoulis et al (Cardiovasc Res 2021) shows culture of rat slices up to 3 days. The latter paper also clearly documents the effect of overloading slices to obtain remodelling in vivo. This paper uses a much more sophisticated method and it is surprising that the authors mention this paper only superficially.*

Response: We would like to direct the reviewer's attention to the fact that the novelty of our approach is the combination of the 3D mechanical stimulation with hormonal stimulation (T3/Dex) which was not reported in any of the previous systems. The systems highlighted by the reviewers used uniaxial mechanical stimulation with basic media (M199/ITS). As shown in our data it is only the combination between the 3D mechanical stimulation and T3/Dex that resulted in prolonged culture beyond 3-6 days' time period as previously reported with complete preservation of the heart slice. We believe this is a significant advancement in the field as acknowledged by the other 2 reviewers. We are sorry that the reviewer felt that we didn't sufficiently discuss the previous systems, therefore, we added detailed discussions of the previous systems and clearly described how our new system is an advancement over the previously reported systems (Lines 81-100).

Comment 3: *Novelty of mechanical stimulation: this concept has been exploited clearly in the*

Fischer and Watson paper and optimised in the Pitoulis paper where an accurate recapitulation of the in vivo force-length relationship of the working myocardium is incorporated in the model. The current paper only offers a rudimentary and superficial approximation.

Response: We very respectfully disagree with the comment “*The current paper only offers a rudimentary and superficial approximation*”. We would like to direct the reviewer’s attention to that the system in Fischer, Watson and Pitoulis manuscripts, were using uniaxial mechanical stimulation with basal media (M199/ITS). However, the current manuscript describes a combination of a full 3D mechanical stimulation where the tissue is exposed to fluidic pressures to introduce physiological mechanical stretch rather than pulling them from the 2 ends. We believe that the 3D mechanical stimulation has a major advantage over the uniaxial mechanical stimulation as the muscle fiber directions of the heart are hard to determine and the mechanical stretches in the heart occur in a 3D manner similar to our system. In addition, according to our data that mechanical stimulation alone was not efficient in maintaining the heart slice viability and it needs to be combined with the hormonal stimulation (T3/Dex). For the general reader, we have added this paragraph to the discussion section (Lines 439-470).

Comment 4: The method of simulating mechanical load is not physiological: in the working myocardium, luminal pressures (afterload) increase during systole and decrease during diastole. In the method presented here, pressures are induced in the opposite direction. Can the authors supply a plot of chamber pressure and slice shortening (volume) and compare it to a physiological human pressure -volume loop of the LV?

Response: We would like to thank the reviewer for bringing up this point. We would like to clarify that the aim of the pressures is to introduce the physiological stretches during diastole (full-length stretch) and systole

(contracted length during electrical stimulation) rather than to introduce preload or afterload pressures as you would notice the pressures inside the fluid chamber are around 20mmHg and are changing by 4mmHg during the cardiac cycle (Fig 2A). In addition, we oversized the slices by 25% which is similar to fractional shortening of the heart during systole (Fig. 2B). As you would notice from the figure in the main manuscript that during systole, the cardiac tissue stretches down by 25% and retrieve that stretch back upon diastole. As requested by the reviewers we have created the PV loop for our system (Response to reviewers Fig. 1).

However, we don’t think this would add value to the manuscript, so we didn’t include it in the manuscript. For the general reader, we have clarified our concept under the limitation of the study (Lines 497-502).

Comment 5: Given the above and the observation that Dex and T3 only partially prevent

remodeling (Supplementary Fig 2) it is surprising that the transcriptomic profile was completely unchanged in the MT group. This result is unexpected and requires validation, at least for selected genes, using rtPCR and for protein levels using Western blotting.

Response: We fully agree with the reviewer regarding the importance of the validation for the RNAseq data. We have conducted qRT-PCR for several cardiac and fibroblast genes which have similar expression levels between fresh tissue and MT day 12 group (Supplementary Fig. 7C). In addition, we confirmed some of the differentially expressed genes between fresh tissue and MT day 12 group using qRT-PCR (Supplementary Fig. 7B). For the protein levels, I hope the reviewer appreciates the difficulty of performing western blotting due to the small size of the tissues. However, we performed immunostaining for troponin-T and Connexin43 which appears to be structurally similar between the fresh tissue and the MT day 12 group (Fig. 5A).

Comment 6: The strongest limitation of this study, recognized by the authors, is the absence of knowledge about contractility of the slices. It is deceiving to present data for slice movement implying that this is an active phenomenon while it seems to me that this is mostly passive, due to pressure changes in the chambers. How can the authors show that the slices contract at all upon stimulation, nevermind quantify this contraction? If the slices contract regularly, it is also possible that they contract in response to mechano-electrical feedback and not the electrical stimulation. It is imperative that this is documented.

Response: We would like to thank the reviewer for highlighting this limitation. We were working tirelessly on this important limitation, and we have been able to develop a new method to determine the live strain using Shape from Shading algorithm. This enabled us to differentiate between the strain with and without electrical stimulation from the same heart slices (Fig. 2F). The strain with electrical stimulation is 20% higher compared to without electrical stimulation within the moving regions of the slices (R6-9). This indicates the contribution of the electrical stimulation to the contractile function. The strain was fully maintained in culture over the 12-day period in both MC and MT conditions (Fig. 4D).

Comment 7: Staining: I do not understand the pictures for muscle staining. There seem to be vacuoli or round negative structures that do not resemble the structure of the myocardium. In addition, the fiber orientation of the slices seem to be random, with myocytes oriented at different angles even in the same image (Figure 7). This makes quantification of cell size impossible.

Response: We would like to thank the reviewer for highlighting this point, WGA staining usually stains the cell membrane to determine the edges of the cardiomyocytes and to be able to measure the cross-sectional area. To satisfy the reviewer's comment, in the revised manuscript we co-stained with troponin-T to demonstrate the cardiomyocytes' cross-sectional area (Fig. 7B).

Reviewer 2:

Comment 1: Miller and colleagues present a manuscript describing the generation of a culture system for pig heart slices and presumably human heart slices. This work advances on the prior studies from the Mohamed lab by now adding dynamic preload stretch to the slices as well as

supplementing the media with tri-iodo-thyronine and dexamethasone. This results in the ability to maintain slices for up to 12 days in culture with preserved structural integrity, viability and gene expression. This is a significant advance over previous studies which showed preserved viability for a day or a few days. The authors also suggest that overstretching the tissues mimics cardiac hypertrophy. The manuscript has a number of strengths including describing the custom system to maintain the cardiac tissue culture model as well as the striking histological and gene expression results showing preservation of the tissue for 12 days. This represents an important advance for the field.

Response: We would like to thank the reviewer for his careful review of our manuscript.

Comment 2: Is the electrical stimulation capturing the tissue and triggering contraction? How do the authors determine this?

Response: We would like to thank the reviewer for this great comment. In the revised manuscript, we have included strain analysis which is clearly showing the effect of the electrical stimulation on the contraction, see Figures 2F.

Comment 3: Is electrical stimulation necessary for the results? The simpler the system, the better, so could the stretching protocol be adequate to preserve structure and viability?

Response: This is actually a great idea, and we have already tested this but did not show the data in the original submission. The combination of electrical and mechanical stimulation is needed to maintain the viability and the structural integrity of the tissue. None of them alone was able to maintain variability and structural integrity (see Supplementary Fig. 3).

Comment 4: The authors acknowledge that they cannot determine the contraction of the constructs with the constant cycling pressure protocol, but couldn't some functional measurements be made by transiently turning off the pneumatic device and doing electrical stimulation alone? This could provide a contractile functional output.

Response: We would like to thank the reviewer for this comment which is similar to comment 6 from reviewer 1. As we explained above, we have optimized a new method to assess the strain within the culture.

Comment 5: The authors described differences in fibrosis (e.g. Fig. 3C) in the various slice interventions, but this is difficult to understand in an ex vivo slice preparation. Fibrosis typically involves an intact perfused organ with contributions from circulating cells that can provide relevant cytokines as well as phagocytosis and other remodeling factors. How do the authors envision this is possible in a 300 μm slice of tissue?

Response: We fully agree with the reviewers that this culture system lacks an immune response; however, it still contains fibroblasts that respond to stress and injury by differentiating into myofibroblasts which were previously assessed in this heart slice model (1). For the general reader, we included this paragraph (Lines 472-477).

Comment 6: The authors claim of "pathological remodeling ... providing a proof of principle that the CTCM device can be used to induce overload cardiac hypertrophy." seems overstated. Is there really remodeling of the tissue or is this just an acute effect of stretch? It requires more

than BNP and cell size measurements to demonstrate pathological remodeling. Furthermore, pressure overload models that generate cardiac hypertrophy typically result from an increase in afterload, not preload, as this model proposes to use. This caveat should be addressed.

Response: We fully agree with the reviewer that we overstated the claims regarding the pathological remodeling, therefore we adjusted the claims to be overstressing the tissue slice which leads to induction of stretch hypertrophic signaling. Therefore, this model could be useful in studying the stretch-induced hypertrophic signaling (2, 3) in the absence of humoral or neuronal factors (which are lacking in our system). We included some more assessments for the induction of hypertrophic signaling such as the NFATC4 nuclear translocation (Fig. 7C). For the general reader we addressed added the following paragraph to the limitation of the study (Lines 503-508).

Comment 7: Statistical analysis appears appropriate.

Response: We would like to thank the reviewer for his positive comment.

MINOR COMMENTS

Comment 1: Line 53, “Additionally, using of animal models to create pharmacokinetic profiles of drugs is relatively expensive...” I do not understand how this is relevant rationale for the model which does not address pharmacokinetics in any way.

Response: We agree with the reviewer, and we removed this statement.

Comment 2: Line 66, “the immature nature of the hiPS-CMs and loss of the multicellular complexity of the heart tissue...” There really is no loss but rather ‘lack of ...’

Response: We amended the wording as recommended by the reviewer.

Comment 3: Line 115, “The custom PCB...” Please define PCB.

Response: We defined PCB: printed circuit board

Comment 4: Line 128, “When the maximum pressure is reached...” What defines the maximum pressure. It looks like this is a fixed value that is selected by investigators. Please clarify.

Response: We defined the maximum air chamber pressure: 80 mmHg for norm and 140 mmHg for OS

Comment 5: Figure 1B legend does not describe the schematic shown in Figure 1B

Response: We included a description of the schematic in the figure legend

Comment 6: Line 620, “...representative trances...” should be “...representative traces...”

Response: This typo was corrected

Comment 7: Figure 7A Y axis label, consider NT-Pro-BNP Fold of D0 to clearly define what is plotted.

Response: We corrected the axis label.

Reviewer #3:

Comment 1: I have reviewed the manuscript submitted by Dr. Mohamed and colleagues, "Biomimetic Cardiac Tissue Culture Model (CTCM) to Emulate Cardiac Physiology and Pathophysiology ex vivo" (Manuscript Number: COMMSBIO-22-0379-T). The study describe a method and a device to maintain in culture ventricular slices for prolonged periods of time (12 days) without losing the main characteristics of the native tissue. Mohamed and colleagues have determine in their study that electrical and mechanical stimulation, accompanied by a defined optimized culture media, allows to maintain not only viability, but also structural, molecular and metabolic integrity of the tissue for at least 12 days.

A dozen of papers have been published in the last few years by Dr Mohamed's group and others describing different approaches to maintain myocardial tissue in culture (from humans and other species), showing how to improve its similarity with native tissue, or extending the culture period. In this study, the authors take a step forward, and apply for the first time a dynamic mechanical load to a myocardial tissue from a big mammal (pig), and successfully maintain native characteristics for 12 days.

The development of myocardial tissue culture models, representative of the native physiology and pathophysiology of the heart, will be of great value in the future and immediate horizon of cardiovascular research. Consolidation of methods to achieve this is key to extending its use among a wide range of researchers. With this work, Dr. Mohamed's group has gone a step further in the establishment of such models.

Experimental design is appropriate, the manuscript is clearly written, and provides sufficient detail to allow its replication. I congratulate the authors for a hard work.

Response: We would like to thank the reviewer for his careful review of our manuscript.

Comment 2: As outlined by the authors, a major weakness of the study is the lack of contractility studies or electrophysiological characterization (for example employing microelectrode arrays or optical mapping). Even when cellular structure and gene expression appear to be maintained, electrophysiological assessment should be done to decipher conduction alterations, or arrhythmogenic susceptibility. This should be evaluated in future studies.

Response: Thanks for the comment, we have partially addressed this limitation by incorporating strain analysis and we are currently working on incorporating optical mapping.

Comment 3: In the abstract section, in background, authors talk about their previous studies with human and pig hearts, but nowhere in the abstract is it indicated that the current work is done on pig tissue. Please outline that current study is performed using porcine myocardial tissue.

Response: Thanks for the great observation, we clarified the use of porcine tissue in the abstract.

Comment 4: In line 42, authors say that "this is the first proof of concept for the ability of the CTCM to emulate cardiac pathophysiology", but this has already been pointed out by Pitoulis et al, albeit with a lower throughput approach and lesser characterization, as authors outline in the

discussion. So I would suggest lowering this statement.

Response: We fully agree with the reviewer, and we toned down the statement (Lines 41-43).

Comment 5: line 45, “that incorporates physiological mechanical and humoral cues”, it would be better to say electromechanical instead of mechanical.

Response: We edit the wording as suggested.

Comment 6: Introduction section: lines 85-95. It would be helpful for the reader to outline in which species experiments have been done, as mice/rats hearts do not respond the same way as pig/human hearts to some procedures, like cardioplegic arrestment. For example: line 86, “to keep pig and human cardiac tissue slices”, line 89 “to culture heart slices from different mammals for extended period of time”. Line 91, outline this work is done in pigs (not in humans, but not in rodents).

Response: We thank the reviewer for the great comment, and we followed his suggestion.

Comment 7: line 104: indicate approximate transport times, how long hearts have been maintained in cardioplegia solution?

Response: The pig surgery is done in another building which is across the street, so it is usually less than 10 minutes to transport the heart between the 2 buildings.

Comment 8: a novel reader not familiar with the previously published cardiac tissue culture experiments could appreciate a short description of the approach, not only the reference of the paper. A short description like “electrically stimulating the slices maintained with optimized culture medium”.

Response: Thanks for the comment, we included further detail in the Methods section (Lines 138-153).

Comment 9: line 137: Tyrode’s composition is not described in this paper. List the components of the modified Tyrode’s solution employed, or cite a reference of a work with the same Tyrode’s solution. Outline also the temperature at which is used.

Response: We included the Tyrod’s solution composition, and all cutting was done in ice jacket to keep the temperature at 4°C (Lines 146-147).

Comment 10: line 142: specify approximate time needed for processing and if Tyrode’s is cold or at room temperature.

Response: These details are now included (Lines 161-164).

Comment 11: line 145: is culture media pre-oxygenated as in previous works?

Response: Yes this is true, and we added these details.

Comment 12: line 157: it is not clear for me if these experiments are done in the incubator or out of it, at 37°C or at room temperature, with culture medium or with Tyrode’s.

Response: As it is hard to do the imaging in the incubator, for consistency, all the mechanical

stretch assessments were done at room temperature after changing the media and it was conducted in media. We now included these details (Lines 180-181).

Comment 13: line 267: figure 1b exists, but is not explained in figure 1 legend. Indeed, in legends, what is written as 1 B is in fact 1 C, and 1 C is 1D. References in the main text are correct, but you have to explain 1B in the legend, and move the numeration of 1 C and D accordingly.

Response: Thanks for the observation, we corrected the figure legend and added a description for Fig.1b.

Comment 14: line 300, it would be useful for the reader to include here a brief description of what parameters are used to quantify structural integrity (not only the reference for the published work). I think it would be interesting also to show a graph with the quantification of troponin and connexin, additionally to the representative images showed and the graph of the % of similarity calculated with the AI framework.

Response: We included a brief description of the AI approach for structural integrity. Actually, this AI approach takes into consideration the expression intensity and localization of the troponin and connexin 43 (Lines 334-340).

Comment 15: line 302, this is showed in figure 3b, but is not indicated. The same with “figure 3c” in line 304.

Response: This is corrected.

Comment 16: - lines 327-341. Insert the references for “figure 4a” (in line 331 approx.), “figure 4b” (line 332), and “figure 4c” (line 338).

Response: This is corrected

Comment 17: - line 347, this is figure 4d instead of 5D. Line 352, this is figure 4e instead of 5E.

Response: This was corrected

Comment 18: - about RNAseq and transcriptional study: I think it would be interesting to outline the 16 genes differentially expressed in the MT slices, for example in a small table in the supplement, or a graph with supplementary figure 5.

Figure 4e show sarcomeric genes, but this in not clearly indicated in figure 4e legend, where it states “major cardiac gene”.

Response: We added the table with the expression pattern of the 16 differentially expressed genes (**Supplementary Fig. 7A**). As suggested, we corrected the figure legend to sarcomeric genes.

Comment 19: An additional comment regarding RNAseq study: taking into account that electrophysiology of the cultured slices has not been explored in this work, it could be interesting to show a heatmap, similar to the 4e figure, for the major genes of the ion channels involved in action potential. This could add evidence of electrophysiological maintenance in the

MT slices, even when it should be further assessed in future optical mapping or microelectrode array studies.

Response: We included the requested heat map for the ion channels (**Supplementary Fig. 9**).

Comment 20: Typographical errors:

- line 413: evinced -> evidenced

- line 620: trances -> traces

- line 636: full stop before Artificial intelligence

Response: Thanks, we corrected the typos.

References:

1. F. Perbellini *et al.*, Investigation of cardiac fibroblasts using myocardial slices. *Cardiovascular Research* **114**, 77-89 (2018).
2. J. Sadoshima, S. Izumo, The cellular and molecular response of cardiac myocytes to mechanical stress. *Annu Rev Physiol* **59**, 551-571 (1997).
3. N. Frey, H. A. Katus, E. N. Olson, J. A. Hill, Hypertrophy of the heart: a new therapeutic target? *Circulation* **109**, 1580-1589 (2004).

REVIEWERS' COMMENTS:

Reviewer #2 (Remarks to the Author):

The revised manuscript including the new strain analysis address the comments from my initial review well. This technology will enable the field with better large animal and ultimately human cardiac tissue models.

Reviewer #3 (Remarks to the Author):

The authors have satisfactorily addressed all my comments in the revised version. I have no additional comments and I recommend to accept the paper.

REVIEWERS' COMMENTS:

Reviewer #2 (Remarks to the Author):

Comment 1: The revised manuscript including the new strain analysis address the comments from my initial review well. This technology will enable the field with better large animal and ultimately human cardiac tissue models.

Response: We would like to thank the reviewer for his constructive comments that improved the manuscript.

Reviewer #3 (Remarks to the Author):

Comment 1: The authors have satisfactorily addressed all my comments in the revised version. I have no additional comments and I recommend to accept the paper.

Response: We would like to thank the reviewer for his constructive comments that improved the manuscript.